# OmniInput: An Evaluation Framework for Deep Learning Models on Internet-Scale Data

**Weitang Liu**[1]*, **Yuelei Li**[1], **Ying Wai Li**[2]*, **Tianle Wang**[1], **Zihan Wang**[1], **Yi-Zhuang You**[1], **Jingbo Shang**[1]

**[1]University of California, San Diego, [2]Los Alamos National Laboratory**

**Reviewed on OpenReview:** `https://openreview.net/forum?id=SvOYlVa3VK`

## Abstract

Evaluating machine learning models is important yet challenging in many real-world scenarios. Traditional analysis is dataset-driven, that is, models are evaluated on predefined benchmark datasets. However, these datasets can only cover a limited scope, leaving *unanticipated* inputs untested and weaknesses of the model unrevealed. To overcome this problem, we propose OMNIINPUT, a novel approach to evaluate models comprehensively using an input space (i.e. Internet-scale data). Our method entails efficient sampling of the inputs from the model and estimation of its corresponding output distribution, and an innovative way to calculate the model's precision and recall curve from the output distribution with only modest human annotation effort. In our experiments, we first validate the correctness of OMNIINPUT within a small input space where brute-force enumeration is still possible. We then show that OMNIINPUT can quantitatively evaluate more complex models such as language models (various versions of GPT2, OLMo, and DistilBERT) and computer vision models, and can discover and analyze interesting input patterns in an input space.

## 1 Introduction

Traditional ways of evaluating machine learning (ML) models mostly rely on the use of a pre-defined dataset (Dosovitskiy et al., 2021; Tolstikhin et al., 2021; Steiner et al., 2021; Chen et al., 2021; Zhuang et al., 2022; He et al., 2015; Simonyan & Zisserman, 2014; Szegedy et al., 2015; Huang et al., 2017; Zagoruyko & Komodakis, 2016; Liu et al., 2020; Hendrycks & Gimpel, 2016; Hsu et al., 2020; Lee et al., 2017; 2018; Liang et al., 2018; Mohseni et al., 2020; Ren et al., 2019; Cao et al., 2022; Sun & Li, 2022; Rozsa et al., 2016; Miyato et al., 2018; Kurakin et al., 2016; Xie et al., 2019; Madry et al., 2017). Depending on how these datasets are constructed, many of them can only cover a limited scope among all possible inputs. Models tested against pre-defined datasets might generalize poorly to unseen inputs in a real-world scenario; hence the weaknesses of the model remain unnoticed. For example, data collected in a lab environment cannot certify the performance of a real-world environment which tends to have more out-of-distribution data. This problem becomes more severe as large language models (LLMs) have become widely accessible to the public, allowing users to input virtually any type of prompt, potentially causing the models to behave unpredictably. Besides, modern models are believed to perform similarly to humans. We explore an evaluation method that measures if the model predictions and human annotations remain similar regardless of the type of data fed into the model. Models with human-level performance should not make simple mistakes that humans would not make. This requires models to be tested, beyond pre-defined datasets, against a much more diverse and extensive amount of inputs.

To overcome this issue, we propose evaluating models comprehensively using the datasets drawn from Internet-scale data where annotation of each data point is infeasible. We consider a discrete and finite *input space* in this work. For example, the input space of a large language model can be all the sequences

---

*These authors made substantial contributions to this work.

of possible combinations of tokens given a sequence length. Whereas traditional evaluations typically use a small-scale dataset whose data points can be labeled by humans and fed to the model to calculate the values of the evaluation metrics, our framework considers a much larger dataset with extensive large-scale data (input space). In this scenario, brute-force enumeration and annotation of each data point in such input spaces is typically computationally intractable. An alternative approach is to uniformly sample data from the internet and evaluate the model's predictions on those inputs. This uniform sampling strategy is also impractical due to the vast number of potential inputs—most of which the models do not predict with high confidence. They tend to assign high confidence to a small subset of inputs that they are familiar with, such as those from their training distribution. However, evaluation of the model should be able to demonstrate how the model performs as its confidence changes. Therefore, a more computationally efficient approach is needed to approximate the evaluation results that would be otherwise obtained from a long-time uniform sampling over the input space.

To tackle this challenge of comprehensive model evaluation with a very large "dataset" (input space with a large number of data), we propose OMNIINPUT, a novel approach to calculate the precision and recall curve of a model over an input space with modest human annotation effort. The key to estimating the precision and recall when input space enumeration is infeasible lies in the *output distribution* of the model (Liu et al., 2023). The output distribution is the proportion of inputs corresponding to each output value [1], showing the (relative) difference in the number of inputs for different outputs (e.g., 10% of inputs have output values equal to 1, 25% of inputs have output values equal to 2, etc.). Specifically, we first sample the inputs and estimate the output distribution of the model under evaluation. We then calculate the model's precision and recall at different output values using the output distribution and selective annotations, constructing a precision-recall curve over the input space. As shown in Fig. 1, it consists of four steps:

(a) We employ a well-established sampling and histogram reweighting algorithm (Hukushima & Nemoto, 1996; Swendsen & Wang, 1986) to obtain the output distribution $\rho(z)$ of a trained model (where $z$ denotes the output value of the model), and efficiently sample the inputs from different output value bins (e.g., negative-log-likelihood or NLL for language models). In model evaluation, all possible inputs in the input space should be evaluated with equal importance.

(b) We annotate the sampled inputs. Using language models as an example, we rate how likely the inputs are understandable sentences using a score from 0 to 1 for language models. This is in contrast to the traditional dataset-driven evaluation setting where the entire testing dataset is annotated beforehand. Annotations of data in our scheme take place *after* the data are sampled by the model of interest. Because the sampling algorithm reliably selects representatives from all possible inputs given an output value, our *limited* annotations offer meaningful and relevant insights to the model itself.

(c) We compute the precision for each bin as $r(z)$, then estimate the precision and recall at different threshold values $\lambda$. When aggregating the precision across different bins, a weighted average of $r(z)$ by the output distribution $\rho(z)$ is required. See Sec. 2.2 for details.

(d) We finally put together the precision-recall curve for a comprehensive evaluation of the model performance.

OMNIINPUT enables the comparison of different models beyond pre-defined datasets selected from a limited input space. The resulting precision-recall curve can help decide the limit of the model in real-world deployment. A model with a high Area Under the Precision-Recall (AUPR) curve in OMNIINPUT is expected to agree closely with human judgment. As OMNIINPUT evaluates the model by having the test data sampled uniformly from the input space, it mitigates potential human biases introduced by the test data collection process (Luo et al., 2023; Prabhu et al., 2023; Shu et al., 2020; Leclerc et al., 2022).

We first validate the correctness of OMNIINPUT on a model with a small input space where brute-force enumeration is still possible. We then conduct extensive experiments using OMNIINPUT to evaluate different models with a large input space where brute-force enumeration becomes impossible. Specifically, we show

---

[1] For language models, we use the log-perplexity (or negative-log-likelihood) as the output. We do not treat them as language generators but as language modelers, where a good language model should assign higher scores (low perplexity) to inputs that humans see as reasonable and fluent, and low scores (high perplexity) to inputs that are nonsensical or unreasonable. For computer vision models in MNIST, logit can be used as the output value to be sampled from.

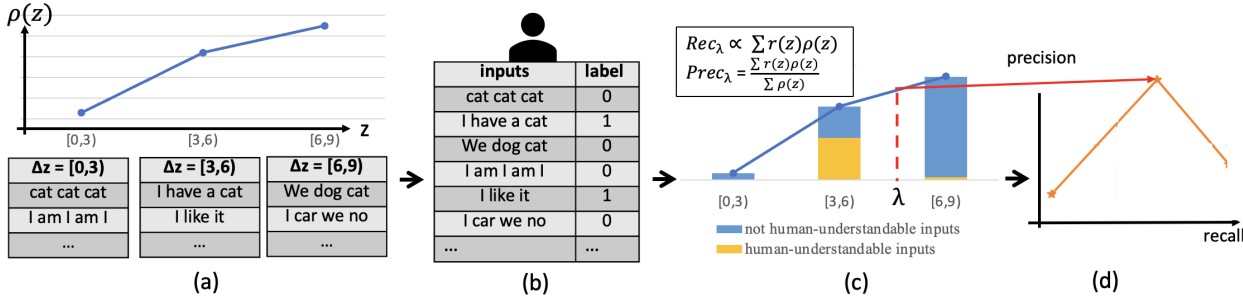

Figure 1: An overview of our novel OMNIINPUT framework for language models as an example. (a) Use an efficient sampler to obtain the output distribution $\rho(z)$ and the sampled inputs for each bin $\Delta z$; (b) Annotate the sampled inputs; (c) Calculate the precision and recall at different thresholds $\lambda$ that distinguish different classes. $r(z)$ denotes the precision of the model within the bin of output value $z$; (d) Construct a precision-recall curve.

that OMNIINPUT applies to (large) language models, such as GPT2 (Radford et al., 2019), OLMo (Groeneveld et al., 2024), and DistilBERT (Sanh, 2019) for text sentiment classification. We also provide an in-depth analysis of computer vision models on the MNIST dataset. All these experiments lead to interesting and sometimes surprising observations. For example, a generative model with a ResNet architecture correctly learns the patterns of the digit "1." However, its recall is significantly lower than that of a simpler multi-layer perceptron classifier, which instead learns "1" by merely inverting the foreground and background of the digit "0." This means the metrics defined in this work (precision and recall) are not indicative of which training method or model architecture is better. Instead, they prioritize other aspects (e.g. reasoning ability, alignment with human intuition, safety, etc) when comparing models. Hence OMNIINPUT, as a proof of concept, paves the way for future research that seeks to better evaluate models by incorporating aspects other than just accuracy. Our new contributions are:

- We propose a novel dataset-free framework, OMNIINPUT, to evaluate machine learning models by estimating the precision-recall curve of the model.
- Our novel evaluation paradigm samples the test data from the model's input space, which goes beyond the use of pre-defined datasets. It includes all possible evaluation perspectives and eliminates possible human biases introduced by the test data collection process.
- OMNIINPUT leverages the output distribution of a model to carefully select a reasonable number of representative samples for humans to annotate. This can reduce human effort in the annotation process, while covering a larger and unbiased input space.
- We empirically validate the correctness of OMNIINPUT and show that it can evaluate various popular (large) language models and computer vision models, and provide detailed analysis.

## 2 The OmniInput Framework

We present the details about sampling the output distribution across the input space, and introduce the OMNIINPUT framework to evaluate model performance by calculating precision and recall in its input space.

### 2.1 Output Distribution and Sampler

**Output Distribution.** We denote a trained neural network as $f : \mathbf{x} \to z$ where $\mathbf{x} \in \Omega_T$ is the training set, $\Omega_T \subseteq \{0, ..., N\}^D$, and $z \in \mathbb{R}$ is the output value that the model assigns to an input. In the example of Natural Language Processing (NLP), $z$ is the NLL which is the log of perplexity. Depending on the task, other outputs could be used. Each of the $D$ tokens takes one of the $N+1$ values. The **output distribution** represents the frequency count of each output $z$ given the (entire) input space $\Omega = \{0, ..., N\}^D$ or some other space $\Omega_M$ specified by a model $M$. The output distribution represents the occurrence for each output value $z$. Following the principle of equal *a priori* probabilities in statistics, we assume that each input within $\Omega$ occurs equally likely. Therefore, similar to the previous evaluations, each input is equally important for analysis and evaluation in our framework. In practice, the statistics of sampling is saved with a histogram

$H(z)$, which stores the occurrence of each output value $z$:

$$H(z) = \sum_{\mathbf{x} \in \Omega} \delta(z - f(\mathbf{x})),$$

where $\delta(\cdot)$ is 1 if $z - f(\mathbf{x})$ is 0, or $\delta(\cdot)$ is 0 otherwise.

The sampled inputs (also called **representative inputs**) predicted with similar output values near $z$ in a small range $[z - \Delta z, z + \Delta z)$ are mapped to the same $z$ bin. $\Delta z$ is a small positive constant chosen empirically. The output distribution $\rho(z)$ is then obtained by normalizing the histogram, that is,

$$\rho(z) = \frac{H(z)}{\sum_z H(z)}. \tag{1}$$

**Discrete Input Sampling and Text Generation.** Markov Chain Monte-Carlo (MCMC) for sampling discrete inputs from ML models are recently proposed (Grathwohl et al., 2021; Zhang et al., 2022). These samplers sample from an ML model to yield a target distribution

$$p_T(\mathbf{x}) \propto \exp(g(\mathbf{x})/T), \tag{2}$$

where $g(\cdot)$ is the negative "energy" and $T$ is a temperature. The sampler initializes a sequence $\mathbf{x}$ of random tokens (or pixels for image inputs). The sampling proceeds by switching some tokens (or pixels) of the sequence to "denoise", and the goal is to acquire the distribution preferred by large $g(\mathbf{x})$. $p_T(\mathbf{x})$ is a quantity dependent on temperature $T$. When $T$ is 1, $g(\cdot)$ becomes the log-probability, a common quantity to be modeled in machine learning (LeCun et al., 2006). $p_T(\mathbf{x})$ in Equ. 2 is also a widely used target distribution in computer vision (Goshvadi et al., 2024) for image generation and natural language processing for text generation (Kumar et al., 2022; Qin et al., 2022).

**Samplers.** Parallel Tempering (PT) (Hukushima & Nemoto, 1996; Swendsen & Wang, 1986) and Histogram Reweighting (HR) (Ferrenberg & Swendsen, 1989) is an efficient approach to obtain output distribution. They are compatible and can take advantage of the development of MCMC samplers that sample from ML models. PT begins with running multiple Metropolis samplers (Metropolis et al., 1953) at different temperatures $T$ simultaneously. The samplers exchange configurations at intervals to accelerate mixing and the thermalization/denoising procedure. Samples are collected by the samplers individually to yield their corresponding target distributions $p_T(\mathbf{x})$. HR then reweighs the sampled distributions and combines them to yield the model's output distribution, which is independent of $T$ and only dependent on the definition of the model itself. Other samplers that sample the output distribution, such as gradient Wang–Landau algorithm Liu et al. (2023), can also be used. However, we adopt PTHR because it is more efficient. As we shall see, we extend the commonly used model evaluation metrics, precision and recall, to an input space. Since precision and recall are summed from high confidence values that correspond to high-quality inputs to low confidence values that correspond to low-quality inputs (low precision), it is not necessary to sample the output distribution for *all* confident values.

## 2.2 Calculation of Precision-Recall

OMNIINPUT revolves around the *output distribution* $\rho(z)$ to formulate the *estimation* of the precision-recall, when combined with additional information from annotated sampled inputs. The samplers described in the previous step provide us with the output distribution, as well as the sampled inputs. For a language model, these sampled inputs are text sequences with $N$ tokens each. We can then proceed to the annotation of these inputs, and use them to calculate the precision and recall.

**Annotation of Inputs.** Our motivation is to quantify how much a model's prediction agrees or deviates from humans' perception, hence humans serve as a gold standard for comparison. For each "bin" of the output distribution (recall that each "bin" collects the inputs with a small range of output values $[z - \Delta z, z + \Delta z)$, a subset of the sampled inputs is selected for annotation and scored by a human. This score ranges from 0 when the sample completely deviates from the annotator's judgment for the target class, to 1 when the prediction from the model perfectly agrees with the annotator's judgment (input with "**good**" prediction).

Following the evaluation, the average score for each bin, termed "precision per bin", $r(z)$, is calculated. It is the proportion of the total evaluation score on the inputs (analogous to true positives) relative to the total number of inputs (analogous to true and false positives) within that bin. The number of bins ranges from 150 to 600, depending on the specific experiments in our study. The mixture interval is a hyperparameter. In our experiments, we simply exchange the configurations every 1000 MC moves.

**Precision and Recall (PR)**[2]**.** In traditional dataset-based model evaluation, precision and recall are widely used to measure the similarity between the model's predictions and ground truths labeled by humans. Humans annotate the inputs based on their understanding of the task they want to achieve. Humans then measure which model's predictions are closer to the ground truth annotations. When the predictions perfectly match the human labels, precision and recall will both be optimal. We further extend the precision and recall framework to the input space of large amounts of data where exhaustive annotation is impossible.

For our experiments on language models, we use the NLL as the output $z$ for sampling, because it is the loss of the next-token prediction. A low NLL corresponds to a human-understandable sentence. We define a varying threshold of model confidence $\lambda$ such that the inputs predicted with $z \leq \lambda$ by the model are similar to their training data, i.e., they are regarded by the model as human-understandable sentences. Thus, the precision given $\lambda$ is defined as

$$\text{precision}_\lambda = \frac{\sum_{z \leq \lambda} r(z)\rho(z)}{\sum_{z \leq \lambda} \rho(z)}. \tag{3}$$

The numerator is the *true positives* which estimates the number of "good" predictions and the denominator is the total number of inputs predicted as positives – the sum of true positives and false positives. This denominator can be interpreted as the *area under curve* (AUC) of the output distribution from the $-\infty$ to threshold $\lambda$. A higher precision indicates a higher proportion of the inputs agreeing with annotators' judgments for the given output values. To compute the recall, we need the total number of ground truth inputs that are positives over the entire input space $\Omega$. This number is a constant, albeit unknown. Hence the recall is proportional to $\sum_{z \leq \lambda} r(z)\rho(z)$:

$$\text{recall}_\lambda = \frac{\sum_{z \leq \lambda} r(z)\rho(z)}{\text{number of positive inputs}} \propto \sum_{z \leq \lambda} r(z)\rho(z). \tag{4}$$

A higher recall indicates that the model makes more predictions that agree with the annotators' (ground truth) judgments. As demonstrated above, the output distribution is a valuable quantity for deriving both precision and unnormalized recall. These metrics can be used to understand the model's mapping by varying the threshold $\lambda$. To simplify the calculation, when $\rho(z)$ differs significantly for different $z$, $\text{precision}_\lambda$ is approximated as $r(z^*)$ where $z^* = \arg\max_{z \geq \lambda} \rho(z)$ and $\text{recall}_\lambda$ is approximately proportional to $\max_{z \geq \lambda} r(z)\rho(z)$.

For the binary classifiers, both (very) negative and (very) positive logit values correspond to the (high) confidence predictions for different classes, separated by the logit=0 which corresponds to probability 0.5 when Sigmoid activation is used. In this case, we need to designate one class, and the lower and upper limits of the summation in Equ. 3 and Equ. 4 should be set accordingly.

**Subsampling of the input space and generative models.** While subsampling the input space seems to be an alternative to OmniInput, it takes a long time to explore the input space thoroughly and find confident samples when the input space contains in-and out-of-distribution (OOD) inputs. To improve efficiency, we propose to reverse the evaluation procedure by first asking the model to find what it believes confidently to be low-confident samples, and humans can annotate the samples selected by the model. The model selects and labels samples much faster than humans do. We show that output distribution is key to estimating precision and recall, the common metrics to evaluate models. Moreover, having a generative model to evaluate another model is an alternative, but we still have to estimate how close this model could generate samples as the original training distribution (huge and finite but not enumerable). Therefore, a large-scale evaluation in an input space for this generative model should go first before using it to test any other models. The generative models trained on internet-scale data are not necessarily able to generate internet-scale data.

---

[2]Another commonly used metric, the Receiver-Operating Characteristic (ROC) curve, is closely connected to PR (see Appendix A).

## 2.3 Application: Model Comparison

One application of OMNIINPUT is to compare models by PR calculated based on each model's own sampled inputs, that reflect the distribution and composition of representative inputs over different output values. Since the precision and recall in OMNIINPUT are computed by the output distributions of the two models and the sampled distributions are represented as histograms, the histograms need to be normalized so that they are comparable. In **Entire output distribution Normalization** we normalize the histograms by dividing the number of sampled inputs (e.g. area under the histogram) so that the area under the curve becomes "1." In this case, it has to sample *all* possible output values and thus covers the space containing $(N+1)^D$ samples. We leverage the fact that the entire input space contains an identical count of $(N+1)^D$ samples for all models under comparison Landau et al. (2004).

However, sampling the entire distribution is not necessary, since the precision and recall curve are summed from high confidence (output) values to low confidence values. We propose **Cross-models normalization** where we need to sample *some* histograms of high confidence values to normalize two models' sampled histograms. We first designate an input subspace from the original input space, such as the subspace whose inputs have their corresponding outputs predicted by both models within a certain range of $\mathbb{Z} = [z_-, z_+]$: $\mathbb{X} = \{\mathbf{x} | M_1(\mathbf{x}) \in \mathbb{Z} \text{ and } M_2(\mathbf{x}) \in \mathbb{Z}\}$. During sampling for $M_1$, we acquire a sampled unnormalized output distribution $\rho_{M_1}(z)$ and the sampled inputs for each $z$. Feeding the sampled inputs from $M_1$ to both models, we find a subset of inputs $\mathbb{X}_{M_1}$ that are from $\mathbb{X}$, meaning that the predicted outputs $z_1, z_2 \in \mathbb{Z}$ for both model when $z_1 = M_1(\mathbf{x})$ and $z_2 = M_2(\mathbf{x})$. We repeat this process for $M_2$ and acquire $|\mathbb{X}_{M_2}|$ as the number of sampled inputs from $M_2$ but these inputs are predicted by both models in $\mathbb{Z}$. We can therefore get the normalized output distributions as

$$\hat{\rho}_{M_1}(z) = \frac{H_{M_1}(z)}{|\mathbb{X}_{M_1}|} \quad (5) \qquad\qquad \hat{\rho}_{M_2}(z) = \frac{H_{M_2}(z)}{|\mathbb{X}_{M_2}|} \quad (6)$$

Both $\hat{\rho}_{M_1}$ and $\hat{\rho}_{M_2}$ are directly comparable, because $\mathbb{X}$ is shared by both of the models. In practice, having $\mathbb{Z}$ is also preferable because not all output values are interesting to consider. For example, a large NLL mostly corresponds to noisy inputs and they generally have very small precision. An intuitive example of applying this normalizaiton is in Appendix B.

**Representative inputs are comparable.** The two models are comparable even though they select different representative inputs. We reason with a hypothetical example. Suppose we pre-train two models to identify human-understandable sentences. After training, we present them with a vast number of sentences from the internet and ask them to select human-understandable ones with 99% confidence. Since labeling every sentence is impractical, we rely on models to perform an efficient selection of human-understandable sentences, which are then annotated by humans. The models examine the sentences extensively (sampling) and then select a few for humans to annotate: one model selects sentences that simply repeat words, while the other selects a mix of human-understandable sentences and sentences containing noisy words. Both models are confident they have chosen human-understandable sentences, but we know they make mistakes.

Despite the differences in representative inputs, we can objectively assess which model identifies more human-understandable sentences, because both models were trained with the same objective and tasked with selecting human-understandable sentences from the same input space. It is important to recognize that the differences in the sentences each model selects directly reflect their distinct abilities, or beliefs, regarding what constitutes a human-understandable sentence. These differences are precisely what OMNIINPUT aims to compare. OMNIINPUT follows a similar approach, but additionally, it requires degrees of uncertainty to compute precision and recall, which necessitates the estimation of the output distribution.

## 3 OmniInput Experiments on Natural Language Processing

**Experimental settings.** We first apply OMNIINPUT to a **Toy** example where enumeration of all inputs is affordable to confirm OMNIINPUT's correctness (Sec. 3.1). In Sec. 3.2, OMNIINPUT is used in two pre-trained GPT2 models (Radford et al., 2019) and OLMo models (Groeneveld et al., 2024) with sequence length 25. We also apply OMNIINPUT to longer sequences of length 100 in GPT-2 models. The sampling target is

the training loss, the negative-log-likelihood (NLL), used for next-token predictions. Finally, we also apply OMNIINPUT to a sentiment classifier.

### 3.1 Toy example validates accuracy of sampling results

**Toy** is a simple GPT2 model with 4 attention heads and 6 layers. We train it to generate a sequence of 8 tokens, each token is an integer between 0 to 9. That is, $\mathbf{x} = \{x_1, x_2, ..., x_8\}$, where $x_i \in \{0, 1, ..., 9\}$. We put a constraint to require the sum of the 8 tokens to be divisible by 30: $(\sum_{i=1}^{8} x_i) \bmod 30 = 0$. Therefore, the input space has $10^8$ inputs, which is enumerable. Our trained model can generate all sequences that satisfy the division-by-30 constraint. Fig 2 shows the resulting $r(z)$ and $\rho(z)$ compared to the ground-truth enumeration, which confirms that they are very close to each other. Since $r(z)$ and $\rho(z)$ are close to the ground truth, the precision and recall estimated through sampling will be close to the ground truth. This provides confidence in extending the approach to more complex applications.

### 3.2 OmniInput evaluation on real-world (large) language models

We apply OMNIINPUT to pretrained GPT-2 and OLMo models to evaluate their performance. Specifically, we test two versions of GPT-2: GPT-2 medium with 25 tokens, referred to as GPT2-medium-25, and GPT-2 small with 25 tokens, referred to as GPT2-small-25. To examine these models' performance on longer sequences, we also test them on 100 tokens, denoted as GPT2-medium-100 and GPT2-

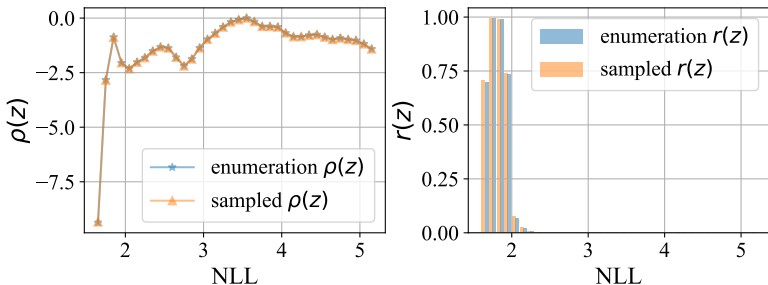

Figure 2: Toy example where enumeration is affordable. The line and bar plots show our sampling results match the ground truth for both output distribution $\rho(z)$ and precision per bin $r(z)$, respectively.

small-100, respectively. For larger models, we apply OMNIINPUT to four pretrained OLMo models with 25 tokens, OLMo-7B, OLMo-1B, OLMo-7B-SFT, and OLMo-7B-Instruct. We use the default vocabulary sizes of 50,304 for OLMo models and 50,257 for GPT-2 models. We emphasize that a test set is not required, as OMNIINPUT does not rely on a predefined dataset for evaluation. Unlike the controlled toy experiments, enumeration is neither assumed nor feasible in real-world applications.

We sample the inputs for different NLL values. A low NLL indicates that the model strongly believes that the sequence is similar to the training distribution (e.g. human-understandable sentences). However, it was found that sequences with NLL below a certain threshold start to contain repeating words and are difficult to be understood by humans (Holtzman et al., 2019). Therefore, we empirically choose an NLL output range and only consider their corresponding inputs when calculating precision and recall (PR). We select the lower bound of the NLL value when it is associated with at least 30 non-duplicate input sequences; the upper bound is constrained by our available human annotation resources. The valid NLL range varies for different models, so we label the inputs with different NLL ranges in different settings. In the GPT2-small-25 and GPT2-medium-25 experiments, we label the inputs with NLL ranging from 2.0 to 4.0. For GPT2-small-100 and GPT2-medium-100, we annotate the outputs with NLL ranging from 4.0 to 5.0. For OLMo models, the NLL range for OLMo-1B is 3.2 to 4.1; 3.4 to 4.3 for OLMo-7B, 3.7 to 4.6 for OLMo-7B-Instruct, and 3.5 to 4.4 for OLMo-7B-SFT. Each bin captures $\Delta$NLL = 0.1.

**Precision-Recall (PR) Curves and Model Performance.** Fig. 3 shows that OMNIINPUT can be applied to produce PR curves for different models and sequence lengths. The PR curves for the two GPT2 models with sequence length 25 demonstrate that these settings generally lead to highly understandable sequences, as indicated by the high precision (see Fig. 3(a)). The results suggest that the two models achieve comparable precision and recall for a sequence length of 25.

We also quantitatively compare the area under the PR curve (AUPR) for the (log-scale) recall (x-axis). A larger AUPR means that the model's predictions agree better with human perspective. First we find the common minimum and maximum recall of the two PR curves for GPT2-small-25 and GPT2-medium-25 respectively, and compute the (linear) interpolation of the precision (y-axis) at each recall. We then

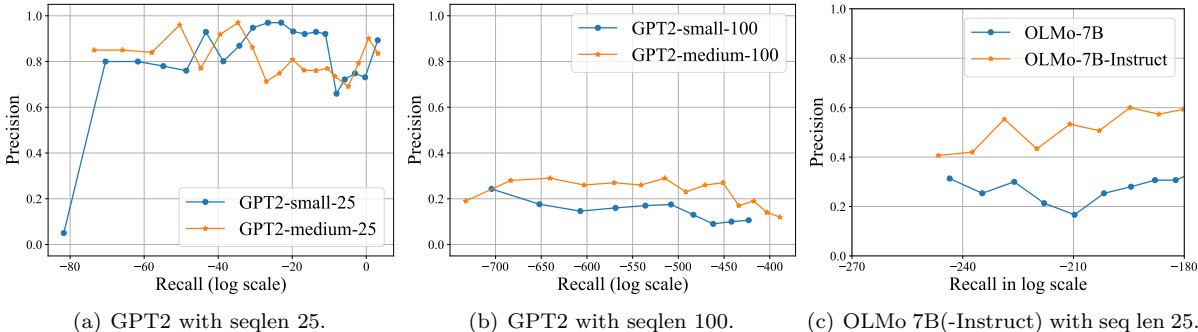

(a) GPT2 with seqlen 25.    (b) GPT2 with seqlen 100.    (c) OLMo 7B(-Instruct) with seq len 25.

Figure 3: Precision-Recall (PR) Curves for Different Language Models.

accumulate the area starting from the common minimum recall and compare the proportion of the AUPRs for the two PR curves. The AUPR of GPT2-medium-25 is then divided by the AUPR of GPT2-small-25. As an example, the proportion of the cumulative AUPR ending at log-recall around 2 (largest recall) is 1.015, indicating the predictions of GPT2-medium-25 agree slightly better with human annotations.

For sequence lengths of 100, OMNIINPUT indicates that GPT2-medium-100 outperforms GPT2-small-100 within the selected output range $\mathbb{Z}$ for the language modeling task. The PR curves of both GPT2 models for sequence length 100 shows that both models generate sequences that are difficult to understand, because of the low precision (see Fig. 3(b)). The PR curve for GPT2-medium-100 is almost always above the PR curve for GPT2-small-100. GPT2-medium has a larger AUPR and thus performs better than GPT2-small for sequence length 100, though both of them have low precision in general.

For OLMo models, OMNIINPUT indicates that OLMo-7B-Instruct outperforms OLMo-7B for sequence lengths of 25 for language modeling task. OLMo-7B-Instruct has a higher precision than OLMo-7B's in a similar recall range, so integrating the PR curve of OLMo-7B-Instruct results in a higher AUPR compared to that of OLMo-7B (see Fig. 3(c)). Therefore, the sequences from OLMo-7B-Instruct are easier to understand. Because the tokenizers are different, we cannot compare across different models.

**Insights from Representative Inputs.** An in-depth analysis by scrutinizing the inputs raises some concerns about privacy leaking and hallucination. For example, an input we encountered contains some company names with their addresses. An internet search on these names or addresses seems to not match, suggesting a potential privacy leak. Another example is a sequence of magic keywords "Good morning dear friend, I, and Greetings, ladies and gentlemen". GPT2-small-25 keeps generating email addresses before this sequence. This raises a concern of privacy leaking of the models.

For GPT2-small-100, we can sample NLL down to around 2.7 where we find repeating words similar to sentences with NLL smaller than 2 in GPT2-small-25. When NLL gets higher, the repeating phrases are generally more meaningful, such as "pickup truck pickup 4 trailer trailer" or "put clean clothes put things wash clothes." Overall, sentences with 100 tokens in the selected NLL range have repeated words and are generally not fluent. More experimental details are in Appendix I. Based on our observations, we speculate that the next-token generation ability of these models, driven by a sophisticated next-token generation function, may not fully align with the NLL for an entire sentence.

**Language classifier.** We fine-tune a DistilBERT (Sanh et al., 2019) model using the SST2 dataset (Socher et al., 2013) and achieve 91% accuracy. We choose the logits as our sampling target, and evaluate this model using OMNIINPUT. OMNIINPUT finds that the language classifier performs poorly, as the precision is 0 for all the recall values. Analysis of the sampled inputs for each positive logit bin reveals that the model tends to classify primarily based on specific keywords, rather than understanding the grammar and structure of the sentences. More details of experiment settings and results are in Appendix D.

Because OMNIINPUT focuses on using PR as the performance metric, it is therefore able to discover the weakness of the model, i.e., disagreement with human's understanding of a meaningful sentence. When the input space contains out-of-distribution (OOD) inputs, it is understandable that the language classifier performs poorly because it is trained to predict the conditional probability $p(class|\mathbf{x})$ given $\mathbf{x}$ are from the training distribution. To handle an open-world setting where OOD's are ubiquitous, the models must also learn to distinguish whether the inputs are from the training distribution.

In summary, our method offers a comprehensive approach to performance assessment. OMNIINPUT is able to capture how well the model's predictions align with human perspective for two reasons: (i) the human annotation process in OMNIINPUT takes place *after* the sampled inputs are determined. The sampled inputs are dissimilar from the training data and have a better chance to reveal "corner cases" or OOD's. (ii) OMNIINPUT evaluates both the precision and recall of the model, which are not used in the loss function during the training procedure. They therefore provide a completely different perspective from the training loss as performance metrics.

# 4 OmniInput Experiments on Computer Vision

We also apply OMNIINPUT to computer vision tasks, demonstrating its versatility across different modalities. In all computer vision experiments, we focus on binary classifiers and energy-based generative models, both of which produce scalar outputs. For classifiers, we consider positive output values (positive logits) $\log p(y = 1|\mathbf{x})$, and for energy-based generative models, we consider log-likelihoods (larger output values).

**Experimental settings.** We evaluate several backbone models for computer vision: convolution neural network (**CNN**), multi-layer perceptron network (**MLP**), and **ResNet** (He et al., 2015). The details of the model architectures are provided in Appendix E. We extract only the inputs with labels $\{0, 1\}$ from the MNIST training set to build the classifiers and refer to this reduced training set as **MNIST-0/1**. We also evaluate generative models for ResNet and MLP. To train these generative models, we select only the inputs with **label=1** as **MNIST-1**; inputs with labels other than **label=1** are considered **OOD** inputs. We train these models using different training methods: (1) Using the vanilla binary cross-entropy loss, we train classifiers **CNN-MNIST-0/1** and **MLP-MNIST-0/1**[3] which achieve test accuracy of 97.87% and 99.95%, respectively. (2) Using the binary cross-entropy loss and data augmentation by adding uniform noise with varying levels of severity to the input images, we train **RES-AUG-MNIST-0/1**, **MLP-AUG-MNIST-0/1**, and **CNN-AUG-MNIST-0/1** which achieve test accuracy of 99.95%, 99.91%, and 99.33%, respectively. (3) Using energy-based models that learn by generating inputs, we train **RES-GEN-MNIST-1** and **MLP-GEN-MNIST-1**[4].

## 4.1 Traditional evaluation is not comprehensive

Traditional dataset evaluation depends on the datasets we choose, so it is inherently not comprehensive. Specifically, we use five different test sets for MNIST binary classifiers by fixing the *positive test samples* as the samples in the MNIST test set with label=1, and varying the *negative test samples* in five different ways: (1) the inputs in the MNIST test set with **label=0** (**in-dist**), and **OOD** inputs from other

Table 1: Traditional evaluations: AUPR scores on pre-defined test sets with five different types of negative samples (see for details), leading to inconsistent evaluation results for model ranking.

| Model | in-dist | out-of-distribution (OOD) | | | |
| | MNIST label=0 | Fashion MNIST | Kuzushiji MNIST | EMNIST | QMNIST |
|---|---|---|---|---|---|
| CNN-MNIST-0/1 | 99.81 | 98.87 | 93.93 | 79.42 | 13.84 |
| RES-GEN-MNIST-1 | 99.99 | **100.00** | **99.99** | **99.87** | **16.49** |
| RES-AUG-MNIST-0/1 | **100.00** | 99.11 | 93.93 | 95.10 | 15.69 |
| MLP-MNIST-0/1 | **100.00** | 99.42 | 92.03 | 90.68 | 15.81 |

datasets: (2) **Fashion MNIST** (Xiao et al., 2017), (3) **Kuzushiji MNIST** (Clanuwat et al., 2018), (4) **EMNIST** (Cohen et al., 2017) with the "byclass" split, and (5) **Q-MNIST** (Yadav & Bottou, 2019).

As indicated by the AUPR scores in Table 1, pre-defined test sets like those mentioned above rarely yield consistent model rankings during evaluation. For example, RES-GEN-MNIST-1 performs the best on all the test sets with OOD inputs while only ranked 3 out of 4 on the in-distribution test set. CNN-MNIST-0/1 outperforms MLP-MNIST-0/1 on Kuzushiji MNIST, but it typically performs the worst on other test sets. Additional inconsistent results using other evaluation metrics can be found in Appendix F.

---

[3]RES-MNIST-0/1 is omitted due to reported sampling issues in ResNet (Liu et al., 2023).
[4]CNN-GEN-MNIST-1 cannot learn to generate reasonable images because the model complexity is low.

## 4.2 OmniInput evaluation

Fig. 4 presents a comprehensive precision-recall curve analysis using OMNIINPUT. The results of OMNIINPUT suggest that RES-AUG-MNIST-0/1 is the best model and MLP-MNIST-0/1 is the second best, both having relatively high recall and precision scores. RES-GEN-MNIST-1, as a generative model, displays a low recall but relatively good precision. Notably, CNN-MNIST-0/1 and CNN-AUG-MNIST-0/1 exhibit almost no precision greater than 0, indicating that "hand-written" digits are rare in the representative inputs even when the logit value is large (see Appendix H).

This suggests that these two models are seriously subjected to an overconfident prediction problem where the models map the wrong inputs (e.g. noisy inputs) over-confidently as "hand-written" digits. In OMNIINPUT, similar to the AUPR calculated from a dataset, a model with higher precision is anticipated to reduce susceptibility to overconfident predictions. Furthermore, a model with a higher AUPR is expected to align more closely with the desired input distribution of the training (testing) set.

We observe the difference in ranking with different datasets in Sec. 4.1. This difference originates from the fact that models understand the training distribution differently. Instead of taking exams with a small set of images with correct answers, the models in OMNIINPUT are asked to select what they believe to be "good" images from a large number of images and we humans as teachers evaluate their selection. Thus, OMNIINPUT is not directly comparable with the traditional dataset-based evaluations.

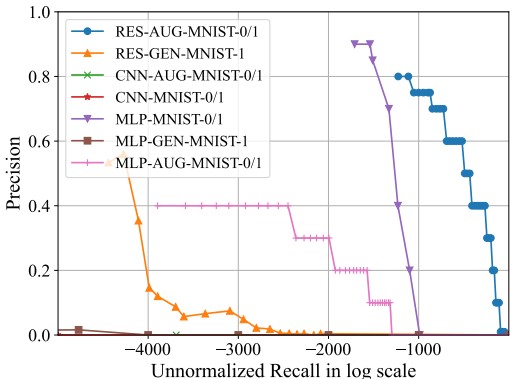

Figure 4: OMNIINPUT evaluation: Precision-Recall curves for various computer vision models. CNN-AUG-MNIST-0/1, CNN-MNIST-0/1, and MLP-MNIST-0/1 have 0 precision.

**Insights from Representative Inputs.** An inspection of the representative inputs (Appendix H) reveals interesting insights. Firstly, different models exhibit distinct input-output mappings and preferences for specific types of inputs, indicating significant variations in their classification criteria. Specifically,

- MLP-MNIST-0/1 and MLP-AUG-MNIST-0/1 likely classify a background-foreground inverted digit "0" as "positive" (digit "1").
- CNN-MNIST-0/1 classifies inputs with a black background as positive.
- RES-GEN-MNIST-1, a generative model, demonstrates that it can map digits to large logit values.
- RES-AUG-MNIST-0/1, a classifier with data augmentation, demonstrates that adding noise during training help the model better map inputs that look like digit "1" to large logit values.

Moreover, RES-AUG-MNIST-0/1 exhibits relatively high recall as the representative inputs generally look like digit "1" with noise in the high logit regime. Conversely, RES-GEN-MNIST-1 generates more visually distinct inputs corresponding to the positive class, but with limited diversity in terms of noise variations. These results suggest that generative training methods can improve the alignment between model predictions and human classification criteria, and underscore the need for enhancing recall in generative models. Adding noise to the data during training can also be beneficial.

**Discussion.** First, although OMNIINPUT would regard CNN-MNIST-0/1 as a "bad" model, the digital "1" can still be found in the positive logit range. The low precision indicates the number of these informative digit inputs is so small that the model makes more overconfident predictions where the models map the obviously wrong instances (e.g. noisy images) over-confidently as "hand-written" digits. Having a mixture of poor-quality and high-quality inputs mapped to the same output indicates a flawed model, so further scrutiny of the inputs is required due to the uninformative and unreliable nature of the model's predictions. Second, the model does not use reliable features, such as "shapes" of the digits patterns to distinguish inputs. Had this model use shapes to achieve high accuracy, the representative inputs would have more similar patterns instead of unstructured or black backgrounds.

We evaluate uninformative samples because an ideal model not only maps training distribution samples to high confidence but also maps the non-training distribution samples to low confidence. It is a common practice in OOD detection to consider a large variety of types of samples, including noisy images and analytical experiments of overconfident OOD prediction on uninformative samples Nguyen et al. (2015) were conducted. The weird failure that the model in these noisy samples that reasonable humans do not make indicates an intrinsic difference in how the model classifies differently than humans do.

The representative inputs of MLP-MNIST-0/1 and MLP-AUG-MNIST-0/1 display visual similarities, but the noise level decreases when the logit increases, indicating how noise affects the model's prediction. OMNIINPUT finds this *distribution shift* in the representative inputs with regard to model outputs. Importantly, this distribution shift is presented by the model itself through the sampling step in OMNIINPUT, without the need for manual trials and errors to find out the noise distribution (Hendrycks & Dietterich, 2019).

Together with the previous PR curve analysis, these findings suggest that different input-output mappings, along with the input distribution, are encoded in the models. This in turn affects the model performance. OMNIINPUT is a novel way to gain insights into this, which can inform future research endeavors to focus on enhancing both the robustness and visual diversity of the generative models.

**Statistics beyond finding OOD inputs.** Besides finding the overconfident OOD samples that humans do not recognize as in-distribution inputs, OMNIINPUT can estimate the *number* of OOD inputs and the number of in-distribution inputs from high (to low) confidence to evaluate models. An optimal model should map a small number of OOD inputs to high-confidence output values. In other words, OMNIINPUT not only finds the individual input as other methods do, such as by adversarial optimization in adversarial attacks but is unique as it can find the the number of them for evaluation. This number is crucial as a better model can still make mistakes, but the number of mistakes is significantly fewer than others.

**Evaluation Effort, Efficiency, and Human Annotation Ambiguity.** We have at least 50 samples per bin for all the models after deleting the duplicates. For a fixed sampling cost, the total number of samples is fixed. Hence a larger number of bins would result in fewer samples per bin. Evaluating these samples in our OMNIINPUT framework requires less effort than annotating a dataset collected for traditional dataset-driven evaluation, e.g., 60000 annotations are needed for the MNIST.

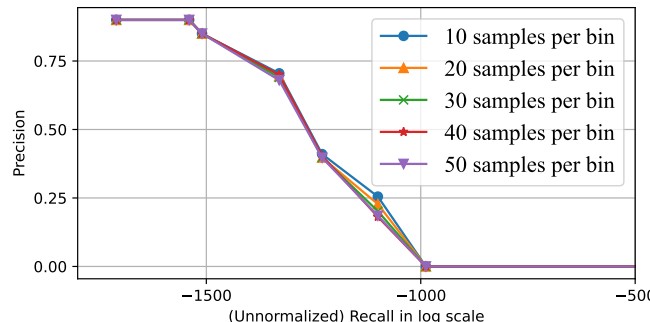

Figure 5: Convergence of OMNIINPUT with respect to the number of annotated samples per bin. We use MLP-MNIST-0/1 as an example. There are 40 bins in total; each bin has at least 50 samples after deleting the duplicates.

In Fig. 5, we vary the number of annotated samples per bin in OMNIINPUT from 10 to 50 and plot different precision-recall curves for the MLP-MNNIST-0/1 model. The results indicate that the evaluation converges quickly when the number of inputs approaches 40 or 50, empirically demonstrating that OMNIINPUT requires relatively few annotated samples to reach a reliable evaluation.

We observe that models exhibit varying degrees of robustness and visual diversity. To assess the ambiguity in human labeling, we examine the variations in $r(z)$ when three different individuals label the same dataset (Fig. 6). Notably, apart from the CNN model, the other models display different levels of labeling ambiguity, i.e., different individuals label the same sample differently. These differences give the confidence intervals in the precision per bin, $r(z)$, and can be understood as the uncertainty of the model predictions.

**Human Annotation vs. Model Annotation.** In principle, metrics employed in evaluating generative models (Salimans et al., 2016; Heusel et al., 2017; Naeem et al., 2020; Sajjadi et al., 2018; Cheema & Urner, 2023) could be employed to obtain the $r(z)$ values in our method. However, using a performance-uncertified model in the entire input space to get $r(z)$ is inherently unreliable. For example, we examined the Fréchet Inception Distance (FID) (Heusel et al., 2017), a commonly used generative model performance metric. We found that the consistency between FID scores and human evaluations varies depending on the models and

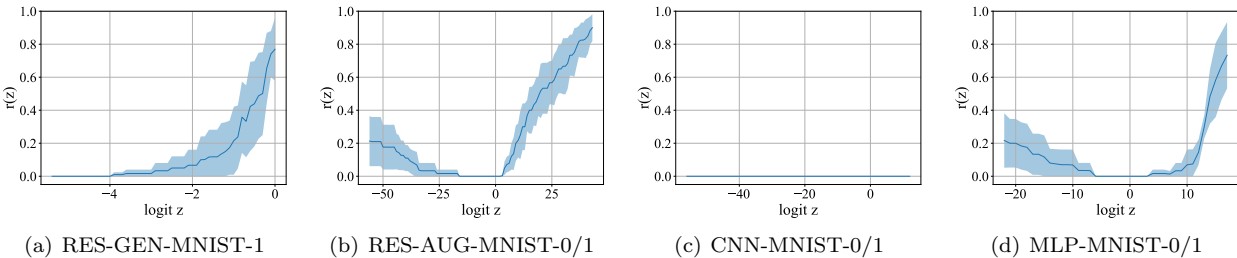

(a) RES-GEN-MNIST-1    (b) RES-AUG-MNIST-0/1    (c) CNN-MNIST-0/1    (d) MLP-MNIST-0/1

Figure 6: Precision per bin $r(z)$ with confidence intervals for four different models.

datasets. This indicates human supervision is needed when model annotation is used. Experiment details are in Appendix G. Currently, model-as-a-judge Gu et al. (2024) is also a widely adopted method in the community, if errors are tolerable. However, when the errors of model annotation cannot be ignored, human annotation is always the gold standard.

## 5 Related Works

**Performance Characterization** has been extensively studied in the literature (Haralick, 1992; Klette et al., 2000; Thacker et al., 2008; Ramesh et al., 1997; Bowyer & Phillips, 1998; Aghdasi, 1994; Ramesh & Haralick, 1992; 1994). Previous research focused on devising evaluation methods for specific model architectures, such as simple models (Hammitt & Bartlett, 1995) and mathematical morphological operators (Gao et al., 2002; Kanungo & Haralick, 1990). In our method, models can be a black box where the analytic characterization of the input-to-output function is unknown (Courtney et al., 1997; Cho et al., 1997), and we place emphasis on the output distribution (Greiffenhagen et al., 2001) to derive performance metrics. This approach allows us to evaluate the model's performance without requiring detailed knowledge of its internal workings. There are recent attempts (Qiu et al., 2020; Lang et al., 2021; Luo et al., 2023; Prabhu et al., 2023) to evaluate model performance without a pre-defined test set. These works used other generators to generate samples for evaluating the model. On the contrary, we use a sampler to sample from the model to be evaluated. Sampling is transparent with convergence estimates, but other generators are still used as black boxes. Because of the inherently unknown biases in generative models, utilizing these models to evaluate another model carries the risk of yielding unfair and potentially incorrect conclusions. Our method brings the focus back to the model to be tested, tasking it with generating samples by itself for scrutiny, rather than relying on external agents such as human or other models to come up with testing data. An additional benefit is that this approach offers a novel framework for estimating errors in the predictions when comparing different models.

**Samplers** MCMC samplers have gained widespread popularity in the machine learning community (Chen et al., 2014; Welling & Teh, 2011; Li et al., 2016; Xu et al., 2018). Among these, CSGLD (Deng et al., 2020) leverages the Wang–Landau algorithm (Wang & Landau, 2001) to comprehensively explore the energy landscape. Gibbs-With-Gradients (GWG)(Grathwohl et al., 2021) extends this approach to discrete domains, while discrete Langevin proposal (DLP)(Zhang et al., 2022) is able to propose global updates. Although these algorithms can in principle be used to sample the output distribution, they either require a long time to converge, or are not able to explore the full range of possible output values in our experience. We therefore employ parallel tempering (PT) with histogram reweighting (HR) to recover the full output distribution of the model, while ensuring a manageable time to solution.

**Open-world Model Evaluation** requires model to perform well for in-distribution test sets (Dosovitskiy et al., 2021; Tolstikhin et al., 2021; Steiner et al., 2021; Chen et al., 2021; Zhuang et al., 2022; He et al., 2015; Simonyan & Zisserman, 2014; Szegedy et al., 2015; Huang et al., 2017; Zagoruyko & Komodakis, 2016), OOD detection (Liu et al., 2020; Hendrycks & Gimpel, 2016; Hendrycks et al., 2019; Hsu et al., 2020; Lee et al., 2017; 2018; Liang et al., 2018; Mohseni et al., 2020; Ren et al., 2019), generalization (Cao et al., 2022; Sun & Li, 2022), and adversarial attacks (Szegedy et al., 2013; Rozsa et al., 2016; Miyato et al., 2018; Kurakin et al., 2016; Xie et al., 2019; Madry et al., 2017). Understanding performance of the model requires the consideration of an input space that includes all these types of samples.

## 6 Conclusions and Future Work

We present OMNIINPUT, a novel framework for realizing dataset-free, comprehensive evaluations of models using Internet-scale data from the models' own input spaces. It is based on a new way to calculate the precision and recall by leveraging the output distribution. We show that OMNIINPUT applies to a wide range of tasks, ranging from computer vision to natural language processing. Our work demonstrates the importance of sampling from the output distribution by showing how it enables the understanding of the model's input-output mapping. Future work can develop more efficient samplers for output distribution, so we can apply OMNIINPUT for evaluating more complex models. Complex models' inference time is longer, so it benefits if more advanced techniques for fast attention calculation are developed. Global updates Zhang et al. (2022) of inputs will lead to faster sampling convergence. It is also interesting to explore how to reduce the required amount of human annotation effort in model evaluations, especially when comparing two correlated models. Continuous input sampling is likely doable but some efforts need to be taken. We believe OMNIINPUT opens the gate to many dataset-free evaluations and this is a crucial research direction, especially in this LLM era when the benchmark dataset performance is less reliable for model comparison.

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

# A  Connection between ROC and PR curve in the input space

It is common in the traditional evaluation framework to consider the receiver operating characteristic curve (ROC) and precision-recall (PR) separately. The recall in PR is the same as the unnormalized true positive rate in ROC, so we do not need to consider the true positive rate separately. The false positive is the number of positively predicted inputs minus the number of the true positives (using the notation of Equ. 3)

$$\sum_{z \geq \lambda}^{+\infty} \rho(z) - \sum_{z \geq \lambda}^{+\infty} r(z)\rho(z) = \sum_{z \geq \lambda}^{+\infty} \rho(z)(1 - r(z))$$

The false positive rate is the number of false positives divided by the number of inputs of the negative class. Since the number of inputs of the negative class is a constant in the input space, the unnormalized false positive rate is:

$$\text{False positive rate} \propto \sum_{z \geq \lambda}^{+\infty} \rho(z)(1 - r(z)).$$

In other words, once we compute the true positive, the false positive rate is simply proportional to the false positive ($\sum_{z \geq \lambda}^{+\infty} \rho(z) - $ true positive) in the input space. Thus, plotting the ROC curve is like plotting $1 - r(z)$ and $r(z)$ scaled by $\rho(z)$ respectively. Comparing the equation of the (unnormalized) recall, this (unnormalized) false positive rate contains (almost if not all) the same information as the (unnormalized) recall in the input space.

# B  Cross-models normalization

As an intuitive example of validation of normalization, suppose it is enumerable to have ground truth count in a large input space where model $M_1$ maps 500 inputs to outputs within $\mathbb{Z}$ and Model $M_2$ maps 200 inputs within $\mathbb{Z}$. $\mathbb{X}$ is the set of inputs that are predicted by both models within $\mathbb{Z}$. $\mathbb{X}$ has 100 inputs.

We sample 50 inputs with outputs within $\mathbb{Z}$ by $M_1$, and it is expected to have around 10 of them from $\mathbb{X}$, because the sampled proportion $\frac{50}{10}$, should be close to the ground truth proportion $\frac{500}{100}$. Therefore, Equ. 5 is $\frac{50}{10} = 5$. We then sample 50 inputs whose outputs are within $\mathbb{Z}$ by $M_2$, and we should have around 25 of them from $\mathbb{X}$. The sampled proportion Equ. 6 is $\frac{50}{25} = 2$, close to the ground truth proportion $\frac{200}{100} = 2$. Therefore, comparing the sampled proportions 5 versus 2 computed from Equ. 5 and Equ. 6 of is as if we were comparing the ground truth count of 500 versus 200.

# C  PR curves for OLMOo models

Fig. 7 shows the PR curves for OLMo-7B, OLMo-7B-Instrct, OLMo-7B-SFT and OLMo-1B.

## D  Language Classifier

We fine-tune a DistilBERT (Sanh et al., 2019) model using the SST2 dataset (Socher et al., 2013) and achieve 91% accuracy. We choose the logits as our sampling target, and evaluate this model using OmniInput. Since the maximum length of SST2 is 66 tokens, one can define the input space for DistilBERT to be sentences with exactly 66 tokens. For shorter sentences, the last few tokens are padded with padding tokens. Because a typical sentence in SST2 contains 10 tokens, we therefore evaluate sentences of 66 tokens in length and 10 tokens in length, respectively.

OmniInput finds that the language classifier performs poorly, as the precision is 0 for all the recall values. Analysis on the sampled inputs for each positive logit bin reveals that the model tends to classify primarily based on specific keywords, rather than understanding the grammar and structure of the sentences.

Here are some sampled inputs for different sentence lengths. For sentence length 66, some sampled inputs with logit equals 7 (positive sentiment) in Fig 8. For sentence length 10, some sampled inputs with logit equals 7 (positive sentiment) in Fig 9.

## E  Details of the Computer Vision Models used in Evaluation

The ResNet used in our experiments is the same as the one used in GWG (Grathwohl et al., 2021). For the input pixels, we employ one-hot encoding and transform them into a 3-channel output through a 3-by-3 convolutional layer. The resulting output is then processed by the backbone models to generate features. The CNN backbone consists of two 2-layer 3-by-3 convolutional filters with 32 and 128 output channels, respectively. The MLP backbone comprises a single hidden layer with flattened images as inputs and produced 128-dimensional features as output. All the features from the backbone models are ultimately passed through a fully-connected layer to generate a scalar output.

## F  Traditional Model Evaluation Results

Tab. 2 shows the AUROC of different models based on pre-defined test sets with different negative class(es). The MLP-MNIST-0/1 performs better on Fashion MNIST but worse in the rest than RES-AUG-MNIST-0/1. RES-GEN-MNIST-1 usually perform the best. CNN-MNIST-0/1 performs better in Kuzushiji MNIST than RES-AUG-MNIST-0/1 and MLP-MNIST-0/1 but worse on the rest. Tab 3 shows the FPR95 results. CNN-MNIST-0/1 performs better on Kuzushiji MNIST than RES-AUG-MNIST-0/1 and MLP-MNIST-0/1 but worse on the rest. These results show the inconsistency between the metrics, dataset and the models.

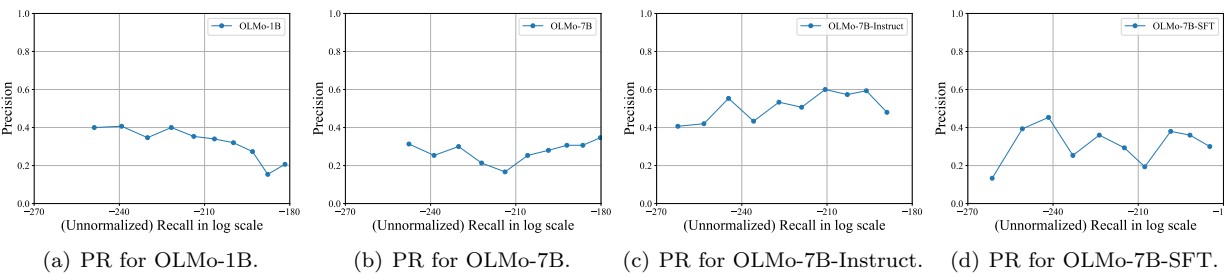

(a) PR for OLMo-1B.  (b) PR for OLMo-7B.  (c) PR for OLMo-7B-Instruct.  (d) PR for OLMo-7B-SFT.

Figure 7: PR for OLMo Language models.

['[CLS]', 'positive', 'dazzling', 'textual', 'quilt', 'shale', 'funk', 'austro', 'advanced', 'tending', 'animals', 'grasping', 'mann', 'scott', 'lower', 'knives', 'avant', 'luckily', '##gui', 'nations', '##48', 'lions', 'nixon', 'steer', 'instituted', 'mont', '##uo', 'hang', 'muir', 'dublin', 'armchair', 'lips', '##tin', 'pianist', 'introduce', '.', 'gunn', 'rosenberg', 'sarawak', 'eddy', '##manship', 'deluxe', 'highway', '##gaard', 'entertain', 'chronic', '##jing', 'objects', 'sw', '##flies', '##tri', 'root', '##phone', 'franciscan', 'longitudinal', 'dealing', 'emilio', 'godfrey', 'audiences', 'comparison', 'shards', 'friendship', 'emphasized', '##ssel', '##ssen', '[SEP]']

['[CLS]', 'positive', 'dazzling', 'textual', 'quilt', 'skill', 'animal', 'fein', 'jocelyn', 'compelling', 'bounce', '##rson', 'mcgraw', 'dynasty', 'buy', '##fight', '##ㄷ', 'republics', 'fictional', '##umble', 'spaniards', 'ronnie', 'wise', 'baha', 'chefs', 'flipping', 'pa', 'symphonies', '##ryn', 'seaman', '##bler', '##ia', '##3', '##tius', 'nests', '.', 'growing', 'phosphorus', 'stakes', '##wski', 'penalty', 'killers', 'manages', '##hue', '##tions', '##rval', 'modify', '##rong', 'bikes', 'frankenstein', 'hayden', 'shirt', 'satisfaction', 'taylor', 'modes', 'audiences', 'impact', '##ska', 'shirley', 'albanians', 'playboy', 'extensions', 'mongolian', 'saturn', '1692', '[SEP]']

['[CLS]', 'positive', 'dazzling', 'textual', 'coloring', 'lays', 'bsc', 'fold', 'michael', 'metre', '332', 'herself', 'von', 'silhouette', 'protestant', 'sonata', 'emblem', 'rag', 'fictional', 'lb', 'yours', 'generator', 'chorale', 'kits', 'marine', '##haya', '##rdes', 'aegean', '350', 'jailed', 'sucks', 'magical', 'graveyard', 'fragile', '##oco', 'hostage', 'honestly', 'retirement', 'wiley', 'interpreted', '"', '##ooping', '##sat', 'devices', 'domesday', 'animation', 'nokia', 'doctoral', 'erich', 'prefix', 'nectar', 'telling', 'wrapping', '##ight', 'herrera', 'fiona', 'stella', 'various', 'since', 'ㄴ', 'arcade', 'passengers', 'terrace', 'newcastle', 'impact', '[SEP]']

['[CLS]', 'positive', 'dazzling', 'textual', 'coloring', 'izzy', 'fisher', 'housing', 'knock', 'supplier', 'park', 'cigar', 'costume', 'essay', 'maple', 'cemetery', 'walton', 'herman', 'like', 'ethernet', 'strikeouts', '花', 'reconstruction', 'distal', '##rien', 'asking', 'choral', 'adventures', '»', 'nucleus', 'accounts', '102', 'illinois', 'is', 'luna', 'hostage', 'clans', 'shit', 'seventeen', '##²', 'canterbury', 'semiconductor', 'childbirth', 'cock', '##iza', 'themed', 'elmer', 'jin', '켜', '##uta', 'cordoba', 'palatine', 'moose', 'dir', 'passenger', 'teller', 'craters', '1710', 'yearbook', 'η', 'jude', 'decades', '##cards', 'santana', '##ume', '[SEP]']

['[CLS]', 'genuinely', 'dazzling', 'textual', 'streaks', '##quist', 'founders', 'generals', 'khan', '##st', 'mahogany', '##evich', 'rwanda', 'penguin', 'bobbed', 'detroit', 'anwar', 'oppression', '##hak', '##isches', 'salmon', '##rien', 'deportation', 'flirt', 'mongolian', '##brush', 'second', 'adventures', 'liquids', 'birth', 'traditional', 'turned', 'induced', 'philharmonic', 'swept', 'stallion', 'geometridae', 'mohan', 'thoughts', '##onga', 'bullock', 'mourning', 'wei', 'teen', 'knighted', 'bavaria', 'atkins', 'peterson', 'ud', 'corona', 'gripped', 'strands', '##iel', 'barclay', 'arranged', 'pune', 'wraps', 'at', 'clues', 'ether', 'strait', 'czechoslovakia', '##rith', 'son', '##glia', '[SEP]']

Figure 8: Sampled inputs of SST2 with sentence length 66.

['[CLS]', 'brave', 'searing', 'vivid', 'nbl', 'restoring', 'uploaded', 'sleeps', 'loyalists', '[SEP]']
['[CLS]', 'appreciated', 'shattering', 'nile', 'barack', 'branch', 'lifelong', 'flavor', 'cow', '[SEP]']
['[CLS]', 'incredibly', 'refreshing', '勝', 'transport', 'teddy', 'fledgling', 'μ', '##pie', '[SEP]']
['[CLS]', 'lexington', 'band', 'difficult', 'prophets', 'humanitarian', 'bianca', 'detectives', 'beautifully', '[SEP]']
['[CLS]', 'made', '##onzo', 'folklore', 'extraordinary', 'islands', 'gameplay', 'absolutely', 'summons', '[SEP]']
['[CLS]', 'generous', 'acceleration', 'precision', '1792', 'freiburg', 'signature', 'treasure', 'parkinson', '[SEP]']
['[CLS]', 'vibrant', 'keen', 'aboriginal', 'psychiatrist', 'scott', 'monumental', '##tical', '1920', '[SEP]']
['[CLS]', 'contraction', 'businessman', 'sunderland', '##away', 'jewelry', 'harmony', 'inspiring', 'realistic', '[SEP]']

Figure 9: Sampled inputs of SST2 with sentence length 10.

| Test Set | class=0 (in-dist) | Fashion MNIST (OOD) | Kuzushiji MNIST (OOD) | EMNIST (OOD) | QMNIST (OOD) |
|---|---|---|---|---|---|
| CNN-MNIST-0/1 | 99.76 | 99.88 | 99.31 | 99.56 | 92.46 |
| RES-GEN-MNIST-1† | 99.99 | 100.00 | 100.00 | 100.00 | 94.85 |
| RES-AUG-MNIST-0/1 | 100.00 | 99.91 | 99.15 | 99.93 | 94.32 |
| MLP-MNIST-0/1 | 100.00 | 99.93 | 98.62 | 99.83 | 94.17 |

†Class=0 is OOD for GEN model.

Table 2: AUROC. The higher the better.

Table 4: Comparing human evaluations and FID scores as metrics of model performance. Although FID scores are similar for the two models, human labels differ significantly from the FID scores. OMNIINPUT evaluates on the input space and thus the the evaluation is more consistent.

| RES-AUG-MNIST-0/1 | | | CNN-MNIST-0/1 | | |
|---|---|---|---|---|---|
| logits | humans↑ | FID↓ | logits | humans↑ | FID↓ |
| 43 | 0.9 | 360.23 | 12 | 0 | 346.42 |
| 42 | 0.88 | 362.82 | 11 | 0 | 358.37 |
| 41 | 0.85 | 368.75 | 10 | 0 | 363.23 |
| 40 | 0.83 | 375.58 | 9 | 0 | 365.01 |

| Test Set | class=0 (in-dist) | Fashion MNIST (OOD) | Kuzushiji MNIST (OOD) | EMNIST (OOD) | QMNIST (OOD) |
|---|---|---|---|---|---|
| CNN-MNIST-0/1 | 0.54 | 0.51 | 2.78 | 1.98 | 21.08 |
| RES-GEN-MNIST-1† | 0.00 | 0.00 | 0.00 | 0.00 | 10.55 |
| RES-AUG-MNIST-0/1 | 0.00 | 0.34 | 4.60 | 0.31 | 14.17 |
| MLP-MNIST-0/1 | 0.00 | 0.27 | 6.68 | 0.64 | 13.24 |

†Class=0 is OOD for GEN model.

Table 3: FPR95. The lower the better.

# G  Human-model annotation inconsistency

Feature extractors generate features from both the ground truth test set images and the images produced by the generative model, and then compares the distributional differences between these features. In our experiment, the ground truth inputs are test set digits from label=1.

The consistency of FID scores with human evaluations varies depending on the models and datasets. For example, in the case of RES-AUG-MNIST-0/1 (Table 4), a decrease in the FID score (indicating better performance) corresponds to an increase in the human score (also indicating better performance) as the logit value rises. This result suggests that certain metrics, like FID, may serve as substitutes for human annotations when evaluating generative models. However, for CNN-MNIST-0/1, the FID score can be entirely misleading. While human evaluators perceive the representative inputs of CNN-MNIST-0/1 for the logits in the table as unmeaningful (having a score of 0), the FID scores are similar to those of RES-AUG-MNIST-0/1, which contains meaningful digits. This is not the only inconsistent case between humans and metrics. For instance, in table 5 of MLP-MNIST-0/1, the FID scores indicate the samples are bad when humans think they are good. The FID scores indicate the even better performance (lower scores) in the logit ranges when humans label as incorrect in general.

When the logits are large, humans label the representative inputs as "1." When the logits are small, representative inputs look like "0." However, the FID scores are better for these "0" inputs, indicating the feature extractors believe these "0" inputs look more like digits "1." The key contradiction is that the feature extractors of these metrics, when trained on certain datasets, are not verified to be applicable to all OOD settings, but surely they will be applied in OOD settings to generate features of inputs from models for evaluation. It is difficult to ensure they will perform reliably.

# H  Representative inputs for MNIST images

Representative inputs for different models are in Fig. 10.

# I  Example sampled inputs of (Large) Language models

We include a few example samples for GPT2 and OlMo in Tab. 6.

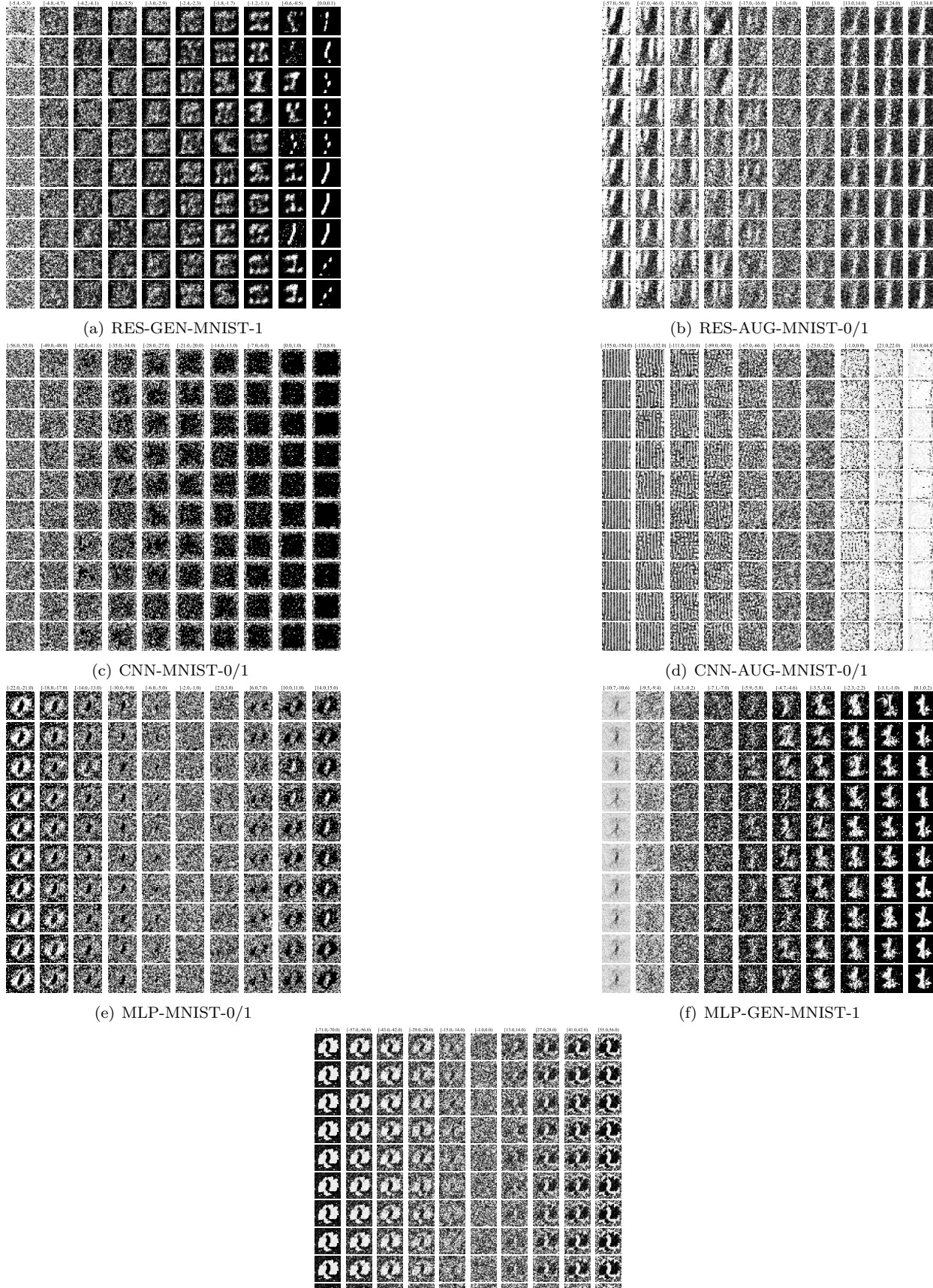

Figure 10: Representative inputs of different models.

| logits | humans↑ | FID↓ |
|---|---|---|
| 17 | 0.73 | 434.32 |
| 16 | 0.67 | 436.60 |
| 15 | 0.58 | 432.89 |
| 14 | 0.48 | 430.79 |
| -19 | 0.18 | 422.01 |
| -20 | 0.2 | 419.94 |
| -21 | 0.2 | 412.96 |
| -22 | 0.216 | 405.20 |

Table 5: For MLP-MNIST-0/1, the FID scores indicate the samples are bad when humans think they are good. The FID scores indicate the even better performance (lower scores) in the logit ranges when humans label as incorrect in general.

| Model | NLL | Samples |
|---|---|---|
| GPT2-small-25 | 1.5 | stores, grocery stores, restaurants, grocery stores, grocery stores, grocery stores, grocery shopping centers, supermarkets, restaurants, grocery |
| GPT2-small-25 | 2.0 | science, science fiction and science-fiction films, Final Fantasy VII, and The Lord of the Rings, among other titles. |
| GPT2-small-25 | 2.5 | Dean was really nice and kind. He was sort of a cuddly. I love him, he's such a good |
| GPT2-small-25 | 3.0 | ethnic tendencies. Students continue to be taught that their current situation is their own fault and they should be taken to task for it |
| GPT2-small-25 | 3.2 | The band responded that they do not refer to themselves as "old school," but rather as modern-era |
| GPT2-small-25 | 3.5 | lure Daphne, Harry. Now our world knows Hermione Granger wasn't the only person in the school who was kidnapped. |
| GPT2-small-25 | 4.0 | Flags and weight fluctuations in the water with power plants and dissipating energy put a lot of strain on the water control system because |
| GPT2-medium-25 | 1.7 | houses, government buildings, schools, hospitals, and other buildings used for public purposes, such as airports, military bases, and |
| GPT2-medium-25 | 2.0 | sadness, but it's a good feeling, and there's a sense that we're better because of it. And that's |
| GPT2-medium-25 | 2.5 | Communists were trying to re-create the experiences of the American Civil War where black soldiers fought alongside white soldiers and played a crucial |
| GPT2-medium-25 | 3.0 | drastically altered. In the early monastic period the belief in eternal life and free will had been transformed into a religious belief in |
| GPT2-medium-25 | 3.2 | acknowledging the wickedness of man due to him, in venting his indignation against him and finally cursing him, he says |
| GPT2-medium-25 | 3.5 | consequence that Chinese law enforcement is very unclear. In fact, given the lack of clear territorial boundaries and with respect to whom the |
| GPT2-medium-25 | 4.0 | disqualifications from Supreme Court proceedings, defence from prosecution, compelment of guardians, obligation to pay retainer of one half |
| OLMo-1B | 3.0 | abouts demeasure on what is to be thereupon upon upon upon upon also also also also also also also too too |
| OLMo-1B | 3.5 | analys anding several changes and adaptations in human brains and their evolution throughout the world history. Brain evolution study discusses how the brain |
| OLMo-1B | 3.8 | farmers traditional ways of life/culture is not acceptable. It comes out from talking which points out the issue of racism, sex |
| OLMo-1B | 4.5 | cis chooses to establish her identity with the Autotaschusansa society," in which typically feminine aspects are a fundamental part |
| OLMo-7B | 3.0 | vp@b v pvphv-p c-a v pvpbv-cpv c-vpv |
| OLMo-7B | 3.5 | wei's C2H probe demonstrated the excellent Ln sensitivity and the low detection limit of Ln. A variety of L |
| OLMo-7B | 3.8 | Whatever happens outcome his ego and character endures it his name will forever thereafter be live in history as it's it's the |
| OLMo-7B-Instruct | 4.0 | Devil and Parker Brothers, for poorly made games which often were inaccessible for children, partly due to the strict licensing rules. As |
| OLMo-7B-Instruct | 4.5 | requisite kms which is all about home address details, favourite foods and what sounds psychical. You will be found that you |
| OLMo-7B-SFT | 3.5 | memory models more often tend towards simple models that treat memory capacity as uniform across all types of memory content. RY: I |
| OLMo-7B-SFT | 4.0 | Something people including me can do.! Creative and idea generating workshops can help you when you need something new for a project |

Table 6: Example Samples

