# OpenReview forum: "OmniInput: An Evaluation Framework for Deep Learning Models on Internet-Scale Data"
_TMLR — Accepted by TMLR_

### Review · Reviewer_Yg7Z · 2025-03-21

**Summary Of Contributions:**

OmniInput is a data-free method for evaluating deep learning classification models. The paper proposes using output sampling (ie sampling stratified by predicted class probabilities) to generate input features, then using those generated inputs to evaluate the model. In particular, the paper proposes to create precision-recall curves from the generated inputs. The proposed approach is applied to the language modeling (ie next-token classification) and image classification settings.

**Audience:**

No

**Claims And Evidence:**

No

**Requested Changes:**

Please see comments above (W1-4) and (M1-3).

**Strengths And Weaknesses:**

Strengths
(S1) Scalable methods for explaining and evaluating models will be of interest to the ML researcher and practitioner communities, and to the TMLR audience in particular.

(S2) Using output distribution sampling to avoid dependence on model internals or a labeled holdout set is a novel proposal.


Weaknesses

(W1) Consider a particular classification model, ModelX. Suppose ImageA and ImageB have the same predicted logits from ModelX. Will OmniInput sample ImageA and ImageB with the same weight, without regard to whether ImageA or ImageB is more realistic (and thus more likely to be fed as input to the model)? If so -- and it appears to me to be so -- this is a major drawback of the proposed method, which limits its usefulness in scenarios where data lives along a restricted manifold of the entire input space. (This is true of the vast majority of common ML scenarios, including in text and computer vision.)

(W2) The paper claims that the vastness of internet data makes it harder to evaluate models, but this seems wrong. The vastness of the internet gives us access to cheap data that we can use to evaluate models (either a selected subset of, or samples from a generative model trained on internet-scale data, for scalability reasons). It's not clear why one would like to use OmniInput's synthetically generated samples in such a scenario. Does OmniInput discover input features that are likely to be adversarial examples for the given model? Does it tend to produce samples on which the model is over-confident, or on which it is under-confident? The paper does not really explain why OmniInput will (allegedly) produce inputs with qualities that make them meaningful or useful for evaluation.

(W3) The empirical results are not very encouraging in this regard. For example, the generated MNIST examples (Figure 10) are mostly noise, or a combination of noise + a particular digit. They're not even diverse with respect to the underlying digit, or following the manifold structure of MNIST inputs. The paper calls these generated samples "representative inputs", but these
samples don't appear to be representative of the MNIST training data. So OmniInput seems to not produce particularly interesting or important examples for evaluating a model. Why should we then care about the AUPR curve evaluated on such uninformative samples?

(W4) The method proposed in "Section 2.3 Application: Model Comparison" either does not make sense, or it is not motivated / explained well, or I am missing something obvious. What exactly is the connection between model comparison (the title of the section) and normalization (which is the focus of the section itself)? How is normalizing an unnormalized distribution sufficient to compare two different models? What is the relationship between "entire output distribution normalization" and "cross-models normalization" -- is the latter an approximation of the former? And does it even make sense to try to compare the AUPR of the two different models when they are being evaluated on each of their own selected "representative inputs"? What if M1 is poorly calibrated compared to M2, so OmniInput generates easy-to-predict inputs on it, inflating its AUPR compared to M2?

Additional comments

(M1) [EDIT: satisfactorily addressed by authors]  ~Re: "A larger AUPR means that the model’s predictions agree better with human perspective." This does not seem true to me. Can you justify this claim?~

(M2) This is perhaps a bit nitpicky, but "We first apply OmniInput to a Toy example where enumeration of all inputs is affordable to confirm OmniInput’s correctness." is not correct. This experiment confirmed that the sampling procedure does in fact correctly generate from the output distribution, but not the correctness of OmniInput (ie the proposal to use the output distribution).

(M3) Maybe I missed this, but I don't see a precise description of the sampling method used. For example, I don't see more details about "The samplers exchange configurations at intervals to accelerate mixing and the thermalization/denoising procedure" -- what are the intervals used? Did these hyperparameters require tuning for the different experiments?

---

> ### Author Response · Authors · 2025-03-24
> **Thank you for the valuable feedback.**
>
> "(W1) Will OmniInput sample ImageA and ImageB with the same weight, without regard to whether ImageA or ImageB is more realistic (and thus more likely to be fed as input to the model)?"
>
> The model will sample ImageA and ImageB with the same weight. If the model maps a realistic looking imageA to high confidence and the bad looking imageB to low confidence, this model is optimal. An ideal model not only maps training distribution samples to high confidence, but maps the non-training distribution samples to low confidence. Application wise, we agree that many common ML scenarios will have semantic meaningful samples. However, in OOD detection, it is a very common practice to use [a large variety of different types of samples](https://github.com/wetliu/energy_ood/blob/master/CIFAR/test.py), including noisy images, to test models. In fact, it is important and typical to test **all possible samples** because intrinsically OOD detection is data agnostic as any data could be the input during testing. Analytical experiments of overconfident OOD prediction have been explored in a space including realistic and non-relistic images [1]. Beyond simply discovering this phenomenon in [1]'s setting, OmniInput seeks to measure the difference between human annotations and model predictions. This setting is intrinsically motivated by the fact that most humans simply won’t make such simple and obvious mistakes on non-realistic looking OOD images, but the models will. Our goal is to build human-like models. Therefore, it is not the data we tested that matters, but whether the model can predict similarly as humans can no matter what data we feed into the model.
>
> (W2) “The paper claims that the vastness of internet data makes it harder to evaluate models, but this seems wrong. The vastness of the internet gives us access to cheap data that we can use to evaluate models (either a selected subset of, or samples from a generative model trained on internet-scale data, for scalability reasons). It's not clear why one would like to use OmniInput's synthetically generated samples in such a scenario.”
>
> Subsampling is one of the approaches, but the samples to evaluate the models are still limited, as human annotations are expensive. In order to make it more efficient, we propose to reverse the evaluation procedure by first asking the model to find by generation from what it believes to be very confident to low confident samples, and human annotation can annotate the samples selected by the model. The model selects and labels samples much faster than humans do. In terms of metrics, we show that output distribution is a key to estimating precision and recall. Moreover, having a generative model is also doable, but we still have to estimate how close this model could generate samples as the original training distribution (huge and finite but not enumerable). Therefore, a large-scale of evaluation in an input space for this generative model should go first before using it to test the other models. Otherwise, these generative models trained on internet-scale data are not certified to generate internet-scale data. OmniInput’s novelty is to make it applicable to compute the metric for input space instead of only for the annotated datasets.
>
> (W2) "Does OmniInput discover input features that are likely to be adversarial examples for the given model? Does it tend to produce samples on which the model is over-confident, or on which it is under-confident?”
>
> As OmniInput also shares the setting of overconfident predictions [1] where the input space also includes noisy samples, it can be viewed as finding adversarial samples without the norm constraints of closeness to the individual test set/training set sample. This delivers a much more comprehensive understanding of adversarial samples which are also overconfident samples including noisy samples. As the sampling target is the log-probability (Fig2(a)), OmniInput will find both the overconfident and underconfident samples.
>
> (W2) "The paper does not explain why OmniInput will (allegedly) produce inputs with qualities that make them meaningful or useful for evaluation."
>
> In OOD detection, it is a very common practice to use various types of samples, including noisy images, to test models. It is important and typical to test **all possible samples** because intrinsically OOD detection is data agnostic as any data could be the input during testing. Analytical experiments of overconfident OOD prediction have been explored in space including realistic and non-realistic images [1]. OmniInput’s goal is to build human-like models. Therefore, it is not the data we tested that matters, but whether the model can predict similarly to humans can no matter what data we feed into the model.
>
> [1] Nguyen, A. Yosinski, J. Clune, J. "Deep Neural Networks Are Easily Fooled: High Confidence Predictions for Unrecognizable Images." CVPR 2015.

---

> > ### Comment · Reviewer_Yg7Z · 2025-03-31
> > **Re: Author response**
> >
> > Thanks for your thorough responses.
> >
> > (M3) Your response addressed my concerns; I had previously overlooked the fact that OmniInput utilizes user annotations on its representative inputs. I have updated my review accordingly.
> >
> > (W4) Re: "What does it mean by "M1 is poor calibrated", and what does it mean to generate easy-to-predict inputs?"
> >
> > Here's a speculative story for how poor calibration might potentially lead to better OmniInput performance. My understanding is that OmniInput does stratified sampling of inputs at various levels of confidence. Presumably for most problems only a tiny fraction of inputs are likely to be high confidencce, so OmniInput upweights such inputs compared to those that are predicted to have a uniform distribution over classes. In other words, OmniInput focuses on samples that are predicted with high confidence. Now suppose for MNIST, M1 and M2 behave identically, except on inputs resembling 7 (one of the harder digits to classify correctly). M1 predicts a uniform distribution over digits to any input that remotely resembles a 7. In contrast, M2 tries to predict a 7 sometimes; sometimes it makes a mistake, but at least it doesn't just always give up. The true-7-but-M1-predicted-as-uncertain samples will be lost as a drop in the vast ocean of noise inputs (which also lead to a uniform distribution over digits), so OmniInput won't likely pick these out for annotation. Meanwhile, OmniInput will probably notice M2's false-positive-7s, and recommend these for annotation. So OmniInput will tell us that M1 is better than M2.
> >
> > I am still considering your responses on the other points, and will respond once I have more time. I appreciate your patience.

---

> > > ### Author Response · Authors · 2025-04-06
> > >
> > > Thanks for your constructive discussion. We are dedicated to address your concerns. We are confident they will be resolved before your submission of decision. Please let us know if you have any questions.

---

> ### Author Response · Authors · 2025-03-25
>
> (W3)"The empirical results are not very encouraging in this regard. For example, the generated MNIST examples (Figure 10) are mostly noise, or a combination of noise + a particular digit. They're not even diverse for the underlying digit, or following the manifold structure of MNIST inputs. The paper calls these generated samples "representative inputs", but these samples don't appear to be representative of the MNIST training data."
>
> The fact that the model trained with diverse samples does not mean the model will necessarily consider these digit samples as the most popular patterns, which is something we need to estimate. We used the well-established sampler that assumes the sampling of the samples is equally likely given a sampling target distribution. Thus, the most popular patterns (no matter if it is a combination of noise or mostly noise) are more likely to be found first. Our sampler in the generative model (Fig 10 RES-GEN-MNIST-1) does lead to much more diverse samples than the other non-generative training schemes, which indicates the diverse patterns depend on the specific model, not a drawback of our samples. In practice, finding diverse samples is a big problem in the sampling community, and will leave it as future work.
>
> (W3) “So OmniInput seems to not produce particularly interesting or important examples for evaluating a model. Why should we then care about the AUPR curve evaluated on such uninformative samples?”
>
> The evaluation of uninformative samples is because an ideal model not only maps training distribution samples to high confidence but maps the non-training distribution samples to low confidence. It is a common practice in OOD detection to consider a large variety of types of samples, including noise images. Analytical experiments of overconfident OOD prediction on uninformative samples [1]. The weird failure that the model in these noisy samples that reasonable humans won’t make indicates an intrinsic difference in how the model will classify differently than humans will. It is not the data we tested that matters, but whether the model can predict similarly as humans can no matter what data we feed into the model.
>
> (W4) “What exactly is the connection between model comparison (the title of the section) and normalization (which is the focus of the section itself) How is normalizing an unnormalized distribution sufficient to compare two different models? What is the relationship between "entire output distribution normalization" and "cross-models normalization" -- is the latter an approximation of the former?"
>
> The precision and recall in OmniInput need the output distributions from two models. The output distributions are represented by the histogram sampled. We need to normalize the histograms so that the output distributions are comparable. In general, any histogram representing distributions is not directly comparable: suppose that we have two Gaussian distributions with two different variances, and usually we sample a different number of samples, say 100 vs 300 samples, and build a histogram of the distributions. The histograms are not comparable – normalization of the histograms by the number of samples (e.g. area under the histogram) so that the area under the curve becomes 1. Using the area under the curve is the “entire output distribution normalization.” However, sampling the entire distribution is time-consuming. If we sample with higher weights on high confidence (output) values, we propose to use cross-model normalizations where we normalize by using only a range of the sampled histograms (high confidences) of two models. Because the normalization between the two models only applies to these two models, it is less general than the entire output distribution normalization which normalizes by only using each model’s sampled histogram.

---

> > ### Author Response · Authors · 2025-03-25
> >
> > (W4)“Does it even make sense to try to compare the AUPR of the two different models when they are being evaluated on each of their own selected "representative inputs"?”
> >
> > We reason with a hypothetical example. Suppose we pre-train two models to identify human-understandable sentences. After training, we present them with a vast number of sentences from the internet and ask them to select human-understandable ones with 99% confidence.
> >
> > Since labeling every sentence is impractical, we rely on models to perform an efficient selection of human-understandable sentences, which are then annotated by humans. The models examine the sentences extensively (sampling) and then select a few for annotation: one model selects sentences that simply repeat words, while the other selects a mix of human-understandable sentences and sentences containing noisy words. Most importantly, both models are confident they have chosen human-understandable sentences, but we know they make mistakes.
> >
> > Despite the differences in input selection, we can objectively assess which model identifies more human-understandable sentences. The fact that both models were trained with the same objective and tasked with selecting human-understandable sentences from the same input space ensures a valid basis for comparison. It is important to recognize that the differences in the sentences each model selects directly reflect their distinct abilities, or beliefs, regarding what constitutes a human-understandable sentence. These differences are precisely what OmniInput aims to compare. OmniInput follows a similar approach, but additionally, it requires degrees of uncertainty to compute precision and recall, which necessitates estimating the output distribution.
> >
> > (W4) “What if M1 is poorly calibrated compared to M2, so OmniInput generates easy-to-predict inputs on it, inflating its AUPR compared to M2?”
> >
> > **Question to the reviewer: What does it mean by "M1 is poor calibrated", and what does it mean to generate easy-to-predict inputs? Do you mean model calibration in general?**

---

> ### Author Response · Authors · 2025-03-25
>
> (M1) "'A larger AUPR means that the model’s predictions agree better with human perspective.' This does not seem true to me. Can you justify this claim?"
>
> Precision and recall are widely used to measure the model's prediction and ground truths labeled by humans, even in traditional dataset settings [2, 3]. When humans annotate the samples, humans have already set up ground truth labels. When we say which model is better, we measure which model's predictions are closer to the ground truth annotations which are based on human understanding. When they perfectly match the human labels, precision and recall will both be optimal.
>
> (M2) "This is perhaps a bit nitpicky, but "We first apply OmniInput to a Toy example where enumeration of all inputs is affordable to confirm OmniInput’s correctness." is not correct. This experiment confirmed that the sampling procedure does in fact correctly generate from the output distribution, but not the correctness of OmniInput (ie the proposal to use the output distribution)."
>
> Yes, OmniInput uses output distribution as a tool to estimate precision and recall. As the sampling results are correct, the computation of the precision and recall will be correct as well. That’s what we mean we shall confidently use OmniInput for other experiments. We will edit accordingly.
>
> (M3) "Maybe I missed this, but I don't see a precise description of the sampling method used. For example, I don't see more details about "The samplers exchange configurations at intervals to accelerate mixing and the thermalization/denoising procedure" -- what are the intervals used? Did these hyperparameters require tuning for the different experiments?"
>
> Parallel tempering is the algorithm [4, 5] we used. It begins with running multiple Metropolis samplers at different temperatures T simultaneously. The samplers exchange configurations. The interval is a hyperparameter, but we don’t need to tune it for each of the experiments as long as it makes exchange configurations while the NLL decreases. We simply exchange the configurations every 1000 steps.
>
> [2] Dan Hendrycks and Kevin Gimpel. "A baseline for detecting misclassified and out-of-distribution examples
> in neural networks."
>
> [3] Shiyu Liang, Yixuan Li, and Rayadurgam Srikant. Enhancing the reliability of out-of-distribution image detection in neural networks. In 6th International Conference on Learning Representations, ICLR 2018, 2018.
>
> [4]Koji Hukushima and Koji Nemoto. Exchange monte carlo method and application to spin glass simulations. Journal of the Physical Society of Japan, 65(6):1604–1608, 1996.
>
> [5] Robert H Swendsen and Jian-Sheng Wang. Replica monte carlo simulation of spin-glasses. Physical review
> letters, 57(21):2607, 1986.
>
>
> **Please let us know if you have questions. Thank you for your reviews and questions.**

---

> ### Author Response · Authors · 2025-04-02
> **Thanks for your constructive discussion.**
>
> Thanks for your constructive discussion.
>
> Let's ensure we are on the same page. Is it correct that we can understand your example as "In high confidence region, because M1 only makes correct prediction on its very certain samples and therefore its accuracy is very high, whereas M2 makes less correct predictions (lower accuracy) but likely to capture more diverse ways digits 7. Thus, it is unfair to only compare the this high confidence region?" If so, it is true that we cannot say M1 is better than M2. In fact, OmniInput has captured this subtle performance difference of precision and recall. In this example, M1 is better than M2 in the sense that its precision is better in the high confidence region. In other words, if we observe a sample predicted with high confidence by M1, we can believe this sample is more likely to be 7. However, because as you mentioned, M1 may miss a lot of true-7-but-M1-predicted-as-uncertain samples, indicating M1 has a poor recall. In order to be a good model, it is very important to achieve both at the same time: an ideal model has to learn as many diverse ways of writing digits 7 to high confidence region (recall), and at the same time, it also has to ensure that this high confidence region only contains digits 7 samples only (precision).
>
> If our understanding does not follow yours, would you like to let us know what the task in your example is?

---

### Review · Reviewer_9Fin · 2025-03-21

**Summary Of Contributions:**

The paper presents OmniInput, a novel dataset-free evaluation framework for deep learning models. Unlike traditional dataset-driven approaches, which are limited to fixed datasets, OmniInput evaluates models across the entire input space by sampling internet-scale data. The key contributions of the paper are:
- OmniInput evaluates models across the entire input space, addressing the limitations of benchmark datasets.
- Supports broader model coverage, tested on large language models (e.g., GPT-2, OLMo) and computer vision models (e.g., CNN, ResNet), demonstrating scalability.
- Precision-Recall curves and AUPR enable meaningful comparison beyond accuracy, considering alignment with human judgements.
.

**Audience:**

Yes

**Broader Impact Concerns:**

The paper mentions that some sampled inputs contained sensitive information (e.g., company names and addresses). This raises concerns about potential privacy violations and unintended data leakage from the training process. Safeguards should be introduced to prevent such privacy risks.

Identifying model weaknesses through OmniInput could lead to adversarial misuse, where attackers exploit discovered vulnerabilities to manipulate model outputs. Strategies to protect models from such attacks should be considered.

Sampling from large input spaces and estimating output distributions for complex models could require significant computational resources, contributing to increased carbon emissions and environmental strain.

**Claims And Evidence:**

Yes

**Requested Changes:**

- The sampling process is computationally expensive for large-scale models. Introducing more efficient sampling methods or approximations would enhance scalability and reduce resource requirements.

- Providing additional details on how the sampling strategy can be adapted to different model architectures and input types would make the framework easier to apply in practice.

**Strengths And Weaknesses:**

Strengths:
1. OmniInput introduces a new paradigm for model evaluation that does not rely on predefined datasets, addressing the limitations of traditional benchmark-based evaluation
2. Sampling from the entire input space allows for evaluation of real-world performance and model generalization
3. The framework minimizes the need for extensive human annotation by sampling representative inputs, increasing evaluation efficiency.
4. The PR curves produced by OmniInput demonstrate its ability to evaluate model performance effectively. GPT2-medium outperforms GPT2-small for longer sequences, and OLMo-7B-Instruct shows improved precision over OLMo-7B

Weaknesses:
1. Sampling from large input spaces, especially for complex generative models, can be computationally expensive despite using efficient methods.
2. OmniInput focuses on discrete inputs, making it less applicable to models with continuous input spaces.
3. Although reduced, human annotation is still required, which could be further minimized through automation or self-supervised learning.

---

> ### Author Response · Authors · 2025-03-26
> **Thank you for your valuable feedbacks.**
>
> (W1) “Sampling from large input spaces, especially for complex generative models, can be computationally expensive despite using efficient methods.”
>
> The evaluation in a large input space is a benefit of our method, even though this problem is hard. The intrinsically difficult problem we chose to solve shall not be considered as a weakness of the method. In section 5, samplers, we have mentioned that future samplers that can be used, such as global updates [1].
>
> [1] Ruqi Zhang, Xingchao Liu, and Qiang Liu. A langevin-like sampler for discrete distributions. International Conference on Machine Learning, 2022.
>
> (W2) “OmniInput focuses on discrete inputs, making it less applicable to models with continuous input spaces.”
>
> Since our method focuses more on a discrete sampling that has wide applications in language models, we did not explore continuous input space. Continuous is likely doable but some efforts need to be taken.
>
> (W3) “Although reduced, human annotation is still required, which could be further minimized through automation or self-supervised learning.”
>
> Human annotation is always the gold standard. LLM as a judge is also a widely adopted method in the community[1]. But the errors of model annotation cannot be ignored. We will leave this a future work.
>
> [2] Gu, J., Jiang, X., Shi, Z., Tan, H., Zhai, X., Xu, C., ... & Guo, J. (2024). A survey on llm-as-a-judge. arXiv preprint arXiv:2411.15594.
>
> (requested changes) “The sampling process is computationally expensive for large-scale models. Introducing more efficient sampling methods or approximations would enhance scalability and reduce resource requirements.
> Providing additional details on how the sampling strategy can be adapted to different model architectures and input types would make the framework easier to apply in practice”
>
> We will add in the manuscript more information on how to train large-scale models and leave comments on sampling algorithms in future work.

---

### Review · Reviewer_StkD · 2025-03-31

**Summary Of Contributions:**

The paper proposes a framework for deep learning model evaluation which should cover the complete input space using internet data, where data points are sampled and manually annotated based on the output distribution of the resp. model. Precision and recall can then be calculated at different output values. The main added argument for such a method is that various deep learning models are not only used for their initial training input distribution, but beyond. The authors propose to use parallel tempering and histogram reweighing for estimating the output distribution.

The approach is evaluated for LLMs (GTP-2, OlMo 7B/-instruct) and vision classifiers (MLP, CNN, ResNet) by pairwise comparison of models wrt the precision call curves or negative log likelihood. The results show that the approach can be used to compare models. It is argued for that this is due to the difference of used data sources, the manual annotations are done after sampling and precision/recall are not used as loss during training.
Additionally, the authors show that in a toy setting (where enumeration is computationally feasible) the sampled distribution is sufficiently equal to the actual distribution.

**Audience:**

Yes

**Broader Impact Concerns:**

There are no serious concerns. It might be interesting to think about the licenses of internet data when applying such an approach, but I am sure one can find good data which are open licensed. Otherwise this should be discussed more.

**Claims And Evidence:**

Yes

**Requested Changes:**

Why does one need an improved sampler compared to (Liu et al. 2023)? Please make this clear in the manuscript.

Can the mentioned approach be used to discover out-of-distribution samples? If so, would it perform better than the SOTA? Please make this clear in the manuscript if this is an important use case.

Extending here, while the evaluation is quite exhaustive for the method at hand, it would be interesting to compare it to more methods listed in the related work section. Could you please argue why the evaluation as is, is sufficient or at further experiments?

**Strengths And Weaknesses:**

# Pros
- The approach to sample from the output distribution and to calculate precision/recall is sensible and sufficiently described in the paper.
- The approach / the direction shows significant value, as model evaluation solely based on small validation/test sets is often not representative for actual use of LLMs or vision models.
- The conducted empirical evaluation comprises both language- as well as vision models, showing sufficient breadth in applications.
- The results show that the metric can be used to discover interesting corner cases

# Cons
- While the approach is sensible, it is unclear to me if the used sampling method is better than referenced work. It should be better shown that there is a real need for it. It is listed as core contribution, so it should be made clear
- The (Liu et al. 2023) reference is missing in the related work in the sample section. It is, again, unclear if this approach could have been reused for sampling or not.

---

> ### Author Response · Authors · 2025-04-04
> **Thank you for your valuable feedbacks.**
>
> (con1) “While the approach is sensible, it is unclear to me if the used sampling method is better than referenced work. It should be better shown that there is a real need for it. It is listed as core contribution, so it should be made clear.”
>
> The sampling method PTHR is a very traditional method to get output distribution so it is not our main contribution. Our main contribution is the importance of the output distribution for model evaluation in an input space where enumeration is impossible. Traditional precision and recall can only be used in a dataset where all the data is annotated; output distribution helps compute the precision and recall in an input space. Second, we chose PTHR because it is more efficient. Importantly, we realized precision and recall are always summed from high confidence values, so there might not need to sample the whole output distribution as Liu et al. did.
>
> (con2) "The (Liu et al. 2023) reference is missing in the related work in the sample section. It is, again, unclear if this approach could have been reused for sampling or not."
>
> The sampler in Liu et al. 2023 can be reused. In fact, any sampler that samples output distribution can be used.
>
> **Requested Changes:**
>
> “Why does one need an improved sampler compared to (Liu et al. 2023)? Please make this clear in the manuscript.”
>
> We chose PTHR because it is more efficient since we realized precision and recall are always computed from high confidence values, so there might not need to sample the whole output distribution as Liu et al. did.
>
> “Can the mentioned approach be used to discover out-of-distribution samples? If so, would it perform better than the SOTA? Please make this clear in the manuscript if this is an important use case.”
>
> Whereas humans don’t recognize them as in distribution samples, the overconfident samples found by our method are overconfident OOD samples. Again our method is to not only discover OOD samples, but also estimate the number of OOD samples and the number of in-distribution samples from high (to low) confidence to get an estimate of how good the model performs. For example, for high confidence output values, the number of OOD samples should be as low as possible so that the model can be an ideal model. In other words, finding the individual samples may be done by other methods such as adversarial optimization (in adversarial attacks), but our method is unique as it can find the statistics of these samples (how many of them).
>
> “Extending here, while the evaluation is quite exhaustive for the method at hand, it would be interesting to compare it to more methods listed in the related work section. Could you please argue why the evaluation as is, is sufficient or at further experiments?”
>
> Because of the datasets we tested, we observe the difference in ranking. However, this difference origins from the fact that models understand the training distribution differently, such as digits 0 vs 1. Instead of examining the models with correct answers of images, we try to ask the model to draw images for us and we humans as teachers evaluate their drawing. Essentially, OmniInput is working on another perspective of the evaluation (evaluation by generation), which is not very comparable with the traditional dataset-based evaluations.

---

### Author Response · Authors · 2025-04-11
**Manuscript changes**

Thank you for the valuable feedback on the changes of the manuscript. Here are the changes we have made so far:

**For Reviewer StkD**

In Sec 2.1 “Samplers”, we explained that we use PTHR over previous work (Liu et al. (2023)) because we don’t have to sample the entire output distribution to compare precision and recall.

In Sec 4.2, we added a paragraph “Statistics beyond finding OOD inputs” to explain OmniInput cannot only find OOD inputs, but also find the number of the bad OOD samples. This number is crucial in understanding model performance.

In Sec.4.2 we mentioned “We observe the difference in ranking with different datasets in Sec. 4.1…”, where we explained the key difference between OmniInput and dataset-based model evaluations.

**For reviewer 9Fin**

In conclusion, we added our thought on continuous sampling.

In Sec 4.2, Human Annotation vs. Model Annotation., we have explained that it is not trivial to use model annotation as errors will occur. However, we also added our thought that model annotation is also a choice if errors are tolerable in the application.

In conclusion, we added future work for efficient sampler and how the complex models help speed up the sampling procedure.

**For reviewer Yg7Z**

(W1, W2, W3) It is a concern of why OmniInput considers uninformative samples. At the end of the 1st paragraph in Sec 1, besides the practical consideration, we added that our goal is to build genuinely human-like models. A human-like model should perform like humans do, no matter what types of samples. Therefore, we consider all types of samples. (Besides, as we explained in 4.2, it is also common in research of OOD detection and understanding the OOD generalization to consider uninformative samples).

(W2) In Sec 2.2 we added “Subsampling of the input space and generative models” to explain why OmniInput can efficiently evaluate the models in the input space but other methods cannot. Output distribution can sample all output values, including over-and under-confident samples.

(W3) In “Discussion” of Sec4.2 (“We evaluate uninformative samples because …”), we argued that it is common in research of OOD detection and understanding the OOD generalization to consider uninformative samples. In Sec 4.2 “Discussion,” we have mentioned that why the our samples find uninformative samples.

(W4) In the 1st and 2nd paragraph of Sec 2.3, we explained the relationship output distribution normalization and model comparison. Specifically, we also explain the difference between the two types of normalizations. At the end of Sec 2.3 small section “Representative inputs are comparable,” We added a hypothetical example of why the models are comparable even though the representative inputs are different. We will be happy to add the content of “calibration” once we have more information from the reviewer.

(M1) In “Precision and Recall (PR)” in Sec.2.2, we explained how precision and recall are used to understand human and model agreement.

(M2) At the end of Sec 3.1, we explained how similarity between ground truth and sampled distributions give us confidence to use OmniInput in complex models.

(M3) At the end of the “Annotation of Inputs” in Sec.2.2, we added our sampler exchange intervals.

Furthermore, to save space, we moved the intuitive example of cross-model normalization in Appendix. We moved the contents in Language classifiers to Appendix. We also shortened “Human Annotation vs. Model Annotation” in the main text and moved the details to Appendix.

Please let us know if you have any questions. Thank you!

---

### Comment · Reviewer_Yg7Z · 2025-05-06
**Unable to review**

I am very sorry, but to due to unforeseen personal circumstances, I don't have bandwidth to continue as a reviewer.

---

### Decision · Action_Editor_ZX3R · 2025-07-10

**Recommendation:** Accept with minor revision

**Additional Comments:**

The authors are responsible for incorporating any and all discussions with reviewers into their paper for the camera-ready version. Authors have already made some progress towards this as specified here: https://openreview.net/forum?id=SvOYlVa3VK&noteId=uwBOqJYhPH

**Audience:**

Yes

**Audience Explanation:**

Submission meets criterion, with the topic being relevant to the TMLR audience, and the work quality meeting the required bar (supported by all to reviewers).

**Claims And Evidence:**

Yes

**Claims Explanation:**

Reviewers feel that the work presented by the authors in the original contribution and over the discussion phase constitutes sufficient evidence that the submission is supported by accurate and convincing evidence. Reviewers (9Fin, StkD) felt that the OmniInput method for generating artificial evaluation samples across the entire input space is interesting and compelling; after discussion they felt the evaluation was sufficient to support claims (StkD).